# History bias and its perturbation of the stimulus representation in the macaque prefrontal cortex

Danilo Benozzo[1] , Lorenzo Ferrucci[2,3] , Francesco Ceccarelli[3,4] and Aldo Genovesio[3,4,5]

[1]*Department of Brain and Behavioral Sciences, University of Pavia, Pavia, Italy*
[2]*Department of Wellbeing, Health and Environmental Sustainability, Sapienza University of Rome*
[3]*Department of Physiology and Pharmacology, Sapienza University, Rome, Italy*
[4]*Institute of Biochemistry and Cell Biology (IBBC), National Research Council of Italy (CNR), Rome, Italy*
[5]*Department of Pharmaceutical Sciences, University of Piemonte Orientale, Novara, Italy*

Handling Editors: Richard Carson & Ricci Hannah

The peer review history is available in the Supporting Information section of this article (https://doi.org/10.1113/JP288070#support-information-section).

The Journal of Physiology

**Abstract figure legend** During the execution of a distance discrimination task, we observed an interference from the most recently presented S2 stimulus on the current-trial performance represented in the figure as weight on the arm of a balance. Specifically there was an attraction of the perceived magnitude of the first stimulus S1 towards the magnitude of the previous trial's S2, that is, S2 one trial back. Behavioural data indicated a response bias consistent with this effect showing an underestimation of S1 when the previous S2 had a low magnitude (S2-ward bias >0) and an overestimation

of S1 when the previous S2 had a high magnitude (S2-ward bias >0). Neural data analyses supported the presence of this interference effect: a linear decoder trained to classify the magnitude of the current S1 showed a systematic bias when tested on trials in which the preceding S2 differed from those in the training set. Its predictions were shifted towards the previous S2 values used for its own training.

**Abstract**   Multiple history biases affect our representation of magnitudes, such as time, distance and size. It is not clear whether the previous stimuli interfere with the discrimination process from the moment of stimulus presentation, during working memory retention or even later during the decision-making phase. We used a spatial discrimination task involving two stimuli of different magnitudes, presented sequentially at various distances from the centre. The monkey's task was to select the farthest of them. We showed that the previous stimulus magnitude produced an attractive effect on the current stimulus magnitude and that this effect was stronger when their stimulus features differed. In this case at the neural level we also observed that decoding of the stimulus magnitude achieved the highest accuracy when it matched the magnitude of the preceding stimulus for which the decoder was trained. This indicates that past stimuli can affect magnitude processing already during the stimulus presentation, even before the decision-making process. Interestingly this effect coincided with an 'activity-silent' period, followed by the reactivation of the decoding of the previous stimulus magnitude.

(Received 11 November 2024; accepted after revision 3 February 2026; first published online 5 March 2026)

**Corresponding authors** D. Benozzo: Department of Brain and Behavioral Sciences, University of Pavia, via Forlanini 6, 27100 Pavia, Italy.     Email: danilo.benozzo@unipv.it
A. Genovesio: Department of Pharmaceutical Sciences, University of Piemonte Orientale, Largo Donegani 2/3 – 28100 Novara.     Email: aldo.genovesio@uniupo.it

## Key points

- Previous experience alters how we perceive the world, including the magnitudes of stimuli.
- We show that the magnitude of the previous stimulus exerts an attractive effect on the perceived magnitude of the current stimulus, and unexpectedly, this effect is enhanced when their visual features mismatch.
- It is still debated whether the history effect results from interference with stimulus processing, from its persistence in memory or during the decision-making phase.
- By recording from the monkey's prefrontal cortex, we found that decoding of the first stimulus is facilitated when its magnitude is similar to that of the recent past stimulus, indicating that the influence of the past stimulus begins during stimulus processing.
- The effect of the previous stimulus magnitude on the representation of the first current stimulus was stronger during the period in which the past stimulus was not explicitly decoded (the activity-silent phase) and preceded its reactivation.

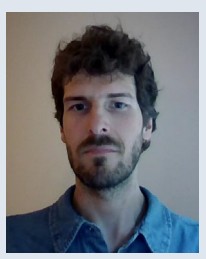

**Danilo Benozzo** received his MSc in bioengineering from the University of Padova (Italy) and his PhD in information and communication technology, with application in neuroscience, from the University of Trento (Italy). He pursued postdoctoral research at both Sapienza University of Rome and the University of Padova, focusing on multivariate analysis and dynamical modelling of neural recordings at both micro- and macro-scales. He is currently a researcher in computational neuroscience at the University of Pavia (Italy).

## Introduction

Previous experience influences and biases our behaviour (Gekas et al., 2019; Maloney et al., 2005). One of these biases is the central tendency bias, first reported by Hollingworth (Hollingworth, 1910), which is also known as the contraction bias (CB). This bias involves the tendency for perceptual estimates to regress towards the mean of the distribution of the set of previously presented stimuli. Evidence of this bias has been described across various domains: in the auditory system (Akrami et al., 2018; Raviv et al., 2014), the visual system (Olkkonen et al., 2014) and the somatosensory system (Fassihi et al., 2017). Additionally it has been observed in the timing domain (Cicchini et al., 2012; Jazayeri and Shadlen, 2010; Roach et al., 2017; Shi and Burr, 2016; Zhang and Zhou, 2017). Interestingly the extent to which this bias alters stimulus perception has been found to be smaller in individuals with dyslexia, in terms of both contraction towards the stimulus mean (during auditory discrimination) and serial dependence (during visual discrimination) (Ahissar et al., 2006; Jaffe-Dax et al., 2016). This has been interpreted as a reduced ability to integrate stimulus statistics, consequently affecting the acquisition of reading skills.

In a controlled laboratory environment, where experimental trials are independent and often predictable, integrating past experiences with current information can introduce a CB, leading to errors influenced by this bias. However when viewed through a Bayesian framework, this CB reveals its advantages, particularly in mitigating the effects of noisy data (Ashourian and Loewenstein, 2011; Bausenhart et al., 2016; Olkkonen et al., 2014; Petzschner et al., 2015; Raviv et al., 2012; Zhang and Zhou, 2017).

CB combines its effect with other serial biases, all originating from previous information's influence on the current trial. Their interconnection is still not completely understood, making it challenging to distinguish the individual contributions of each factor to the overall outcome. Recently Boboeva et al. (2024) suggested a new perspective, proposing that CB is an emergent effect of the interplay between history biases and working memory processing.

An explicit coding of previous information or its impact on neural activity during the current trial is indicative of the effect of previous trials on performance. Several studies have shown the effect of trial history on current neural activity, particularly in relation to previous outcomes and behavioural goals (Barraclough et al., 2004; Genovesio and Ferraina, 2014; Genovesio et al., 2005; Hermoso-Mendizabal et al., 2020; Histed et al., 2009; Seo and Lee, 2009; Seo et al., 2007). Other studies have shown that past experience can not only be represented at the neural level but also bias future choices in the prefrontal cortex, a phenomenon that has been less extensively studied (Barbosa et al., 2020; Mochol et al., 2021; Padoa-Schioppa 2013). Although previous studies have shown that information from previous trials keeps getting encoded in the next trial, it remains unclear whether previous trials can influence the coding of stimulus magnitude information during the stimulus presentation as well, possibly leading to behavioural biases.

Only a few studies have investigated the neural correlates of CB: Akrami et al. (2018) in the rat's parietal cortex, and Benozzo et al. (2023) and Serrano-Fernandez et al. (2024) in the monkey's prefrontal cortex. Through optogenetic inactivation Akrami et al. (2018) showed that the effect of the previous trial was eliminated by switching off the rat's parietal cortex neurons. This translated into an improvement in behavioural performance, showing that, together with their neurophysiological results, at least in rats, the parietal cortex combines current and past information.

We used a two-stimuli distance discrimination task to investigate whether and how the most recent trial interferes with the processing of the current-trial stimuli. In principle information from the previous trial could bias perceptual decisions during any of three task phases: first, during stimulus presentation; second, during the maintenance of the stimulus in memory; or third, during the comparison process when the first and second stimuli are compared. We have previously shown that CB affects the decision process (Benozzo et al., 2023) by biasing the first stimulus magnitude towards the average distribution of stimulus magnitudes. However this did not rule out the possibility that the influence of prior information begins even earlier in the task. In this study we investigated whether the most recent experience, that is the last stimulus presented in the previous trial, can also affect the representation of the current stimulus, thus placing the effect of the previous trial bias at an earlier stage than the decision process.

We found that the current stimulus magnitude could be discriminated with higher accuracy when training and testing were performed on trials with similar previous stimulus magnitudes, pointing to an effect of the past stimulus magnitude on the processing of the stimuli. This effect was not strictly time locked, though it tended to emerge more strongly in the first part of the stimulus presentation, followed by the reactivation of the trace of the past stimulus magnitude, after it was not explicitly coded in the initial part of the stimulus presentation. It remains possible that although not explicitly encoded, previous magnitude information was maintained in the first part of the stimulus in an 'activity-silent' state (Barbosa et al., 2020; Ranieri et al., 2022; Stokes, 2015). An 'activity-silent' state refers to the absence of an explicit coding, not to be confused with the absence of activity. Surprisingly we also discovered that these effects were most pronounced when the visual features of previous

and current stimuli differed, suggesting that the stimulus properties play a triggering role in the influence of prior information.

## Materials and methods

### Animals

Two adult male rhesus monkeys (*Macaca mulatta*) weighing 8.5 (9 years old) and 8.0 kg (11 years old) were used in this study. The monkeys were trained prior to surgery and the start of recordings for a period of ∼2 years, on a Monday to Friday schedule with two resting days during the weekend. To encourage participation during training and neural recordings, the animals' water intake was regulated. Monkeys had unrestricted access to dry food. After each day's experimental sessions they were given fresh foods like fruits and vegetables. Body weight was measured several times per week and maintained at no less than 85% of the pre-water-control baseline weight. Full-time, on-site veterinary personnel closely monitored the animals' weight and overall health. Monkeys were housed in pairs unless temporary separation was necessary due to adverse outcomes. The information on the care and welfare of the animals is the same as that reported in previously published articles that used the same task/animals (Benozzo et al. 2021, 2023; Genovesio et al. 2009, 2011, 2012, 2015, 2016; Londei et al., 2025; Marcos, Tsujimoto et al. 2019; Marcos et al. 2017).

### Behavioural task

In the distance discrimination task two visual stimuli, a blue circle of 3° diameter and a 3° × 3° red square, were consecutively presented on a computer screen in each trial (Fig. 1*A*). Each trial started when the monkeys pressed the central of the three switches, which caused the appearance on the screen of a central stimulus (reference point). After 400 or 800 ms the central stimulus was followed by the onset of S1. Each of the two stimuli, S1 and S2, was presented for 1000 ms at a variable distance (from 8 to 48 mm, in 8 mm steps) either above or below the reference point. After the disappearance of S1 there was a first delay (D1) of 400 or 800 ms before the presentation of S2. S2 appeared above the reference point if S1 had appeared below, and below the reference point otherwise. The presentation of the red square and blue circle of S1 and S2 was independently pseudo-randomized. The distance of S2 never equalled that of S1, and the two stimuli were always different from each other, if S1 was the red square S2 was the blue circle and vice versa. The disappearance of S2 was followed by a second delay (D2) of 0, 400 or 800 ms, which in turn preceded the reappearance of the two stimuli. The reappearance of the two stimuli (horizontally arranged and equally spaced from the reference) represented the GO signal to select the target stimulus previously presented farther from the reference point. Each of the two stimuli was pseudo-randomly chosen to be located either 40 mm to the right or 40 mm to the left of the central stimulus. Because in each trial the blue circle and the red square were pseudo-randomly placed on the left and right sides of the reference point, no (informed) motor response could be planned until the GO signal. A 0.1 mL of fluid reward was delivered after correct selections, whereas an acoustic feedback followed incorrect ones. See Genovesio et al. (2009, 2011, 2012, 2015, 2016), Marcos et al. (2017, 2019) and Benozzo et al. (2021, 2023) for more details on the task.

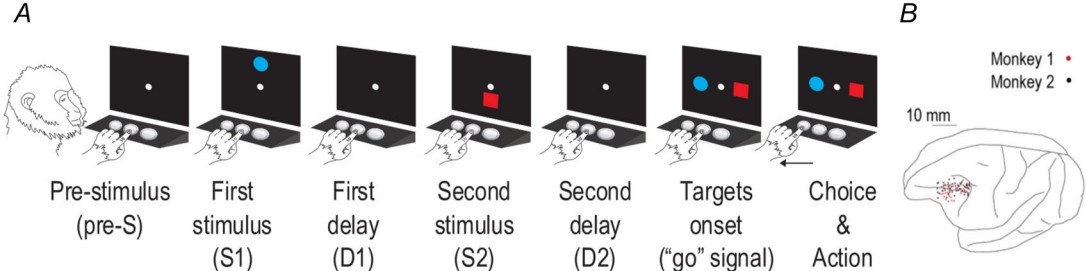

**Figure 1. Experimental task and recording sites**
*A*, trial task events. Trials were initiated by touching the central switch, which led to the appearance of the central stimulus (reference point). After 400 or 800 ms (pre-stimulus, pre-S) S1 was presented. A variable delay (first delay, D1) of 400 or 800 ms followed S1, after which the second stimulus (S2) appeared. S2 was followed by a second delay (second delay, D2) of 0, 400 or 800 ms. Both S1 and S2 were presented for 1000 ms either above or below the reference point by 8–48 mm (8 mm steps). At the target onset both stimuli reappeared horizontally, and the monkey had to select the stimulus that had been presented farther from the reference point (the blue circle in the example trial) by touching the spatially corresponding switch below the screen. Correct responses were rewarded with 0.1 mL fluid, whereas errors were followed by acoustic feedback. The stimulus feature (blue circle/red square), position (above/below the reference point), distance from the reference point and target position (left/right) were pseudo-randomly selected. *B*, penetration sites. Composite from both monkeys, relative to sulcal landmarks.

## Data collection

Data were collected from the dorsolateral prefrontal cortex (PFdl). Eye position was recorded using an infrared oculometer (Arrington Research, Scottsdale, AZ, USA), and single cells were recorded using quartz-insulated, platinum-iridium electrodes (0.5–1.5 MΩ at 1 kHz) and positioned by a 16-electrode drive assembly (Thomas Recording, Giessen, Germany). Single-cell spike trains were sorted using both online and offline sorting algorithms; we used the Multichannel Acquisition Processor (Plexon, Dallas, TX, USA) and the Offline Sorter (Plexon), respectively.

## Surgery and histological analysis

We implanted the recording chambers over the exposed dura mater of the left frontal lobe, with head restraint devices, using aseptic techniques and isofluorane anaesthesia (1%–3%, to effect). Monkey 1 had two 18 mm-diameter chambers, and monkey 2 had a single $27 \times 36$ mm chamber. After surgery the monkeys were monitored daily by full-time on-site veterinary staff who managed the use of therapies and analgesics to ensure full postoperative recovery and the general welfare of the animals. At the end of the experiment, we made electrolytic lesions (15 mA for 10 s, anodal current) at selected locations to localize the cells recorded with more precision using the electrode tracks as landmarks. After 10 days the animal was deeply anaesthetized and perfused through the heart with formaldehyde-containing fixative. The entire killing procedure was conducted under extremely deep anaesthesia and under the supervision of the responsible on-site veterinary staff. We plotted the recording sites on Nissl-stained coronal sections by reference to both the recovered electrolytic lesions and marking pins inserted during perfusion.

## Data analysis

### Analysis of behavioural data

*S2-ward bias.* To assess the influence of the previous trial's S2 magnitude on each stimulus pair (S1, S2), we evaluated the performance for every combination of the current pair and the previous S2. This analysis was performed on the 10 stimulus pairs with a relative distance of 8 mm. Considering that each pair could be preceded by six possible S2 one-trial back magnitudes (ranging from 8 to 48 mm, with 8 mm steps), this resulted in 60 pairwise (($S1,S2$)$_{current}$, $S2_{previous}$) combinations. To enable comparisons between combinations with the same current stimulus pair, we normalized the performance by subtracting the overall mean performance specific to each pair. Our performance metric was the probability of

selecting S2. It is important to note that this probability corresponds to a correct response in cases where S2 > S1, and conversely, it represents the probability of an erroneous response when S2 < S1. When a stimulus pair was considered alongside the preceding S2, we denoted its normalized performance measure as the 'S2-ward bias'. Our decision to focus our analysis only on pairs with a relative stimulus distance of 8 mm was for keeping as much as possible under control the influence of other factors that could contribute to the performance. Every trial included in the analysis followed a complete trial. For testing whether only the latest stimulus information was important, we also performed the same analysis using the magnitude of the preceding S1, which represents the least recent stimulus presented in the prior trial.

*Generalized linear model of behaviour.* A generalized linear model with a binomial distribution and logit link function was used to predict the probability of choosing S2 across stimulus pairs with a relative distance of 8 mm (10 pairs) and their corresponding previous trial pairs (30 previous pairs per current pair). Four combinations of regressors were tested. Two models included only current-trial features based on task difficulty, that is, the stimulus ratio (measured as $sign(S2 - S1) max(S1,S2)/min(S1,S2)$) and the CB effect (measured as the distance of S1 from the true mean stimulus distribution, that is, $S1 - <S>$ where $<S> = 28$ mm). In the other two models we additionally included the magnitude of the previous S2, and, in the last model, also the magnitude of the previous S1. Each model was trained on 300 randomly sampled sessions and tested on another 300 sessions. Predictor values were normalized between 0 and 1, and a positive sign was assigned to difficulty-based regressors when the choice of S2 was favoured. This procedure was repeated 200 times. Model fitting was performed using the Python module statsmodels.

### Analysis of neural data

*Cell and trial selection.* We included only neurons in our analysis with a mean firing rate of at least 1 spk/s, with at least five trials per class, and following a complete trial.

*Population decoding.* To assess the task variable's decoding capability across sessions, we followed the methodology outlined in Rigotti et al. (2013). For each class that we aimed to decode, we created two distinct pseudo-populations. To construct these pseudo-populations we randomly allocated 75% of the trials from each neuron to the training set and reserved the remaining 25% of trials for the test set. We included each recorded neuron in the pseudo-population if it met the criteria for mean firing rate and trial count. To standardize the entire dataset we applied z-score scaling based on the training trials. Subsequently we employed

a linear support vector machine (SVM) classifier with a l2 penalty and a regularization parameter set to 1 to decode each test trial class. We replicated this procedure 100 times to ensure robustness in our analysis. Because our primary focus was on neural activity during the presentation of the first stimulus, we focused on the initial part of the trial. To obtain a decoding curve over time, the spiking rate of each neuron was computed in a 200 ms moving time window, initially centred 300 ms before S1 presentation, and advanced in 25 ms steps until the end of the shorter D1 delay (400 ms). By doing this we stopped 1400 ms after the S1 presentation. In each moving window a new decoder was trained and tested. We formulated the problem as a three-class decoding problem. Considering that the stimulus magnitude ranged from 8 to 48 mm with an 8 mm step size, we grouped the stimulus magnitudes in three classes: low distance (8 and 16 mm), medium distance (24 and 32 mm) and high distance (40 and 48 mm). This choice was motivated by the need to simplify the decoding problem to make it compatible with the limitation of the number of recorded trials. For testing significance a null model was produced by randomly permuting the trial class labels 1000 times. This analysis was performed twice, once considering as trial label the distance of S1 current trial and once the distance of S2 one-trial back.

To study the effect of the history bias, the decoding analysis of S1 in the current trial was repeated conditioning it on the magnitude of S2 one-trial back. Keeping the same stimulus groups, that is, low, medium and high distances, we divided the analysed trials based on their S2 one-trial back magnitude and ran the S1 (current trial) decoding as described before. This implies that we had three different S1 (current trial) decoders that we evaluated both within and across conditions. By within condition we mean that the decoder was trained and evaluated under the same S2 one-trial back condition, whereas across conditions means that the sets of trials in which the decoder was trained and evaluated belonged to different S2 one-trial back groups. From this analysis beyond classification accuracy, we also computed the decoder's marginal probabilities of assigning each of the three classes to a given test trial. This allowed us to detect potential biases in class assignment based on the S2 one-trial back category used during training.

A similar population decoding analysis was also conducted on the visual features of S1 in the current trial. In this experiment all blue stimuli are circles and all red stimuli are squares; therefore colour and shape visual features cannot be dissociated. We will refer only for descriptive purposes to colour differences between stimuli, but the difference can be at the shape level as well. This analysis was motivated by the observed role of stimulus colour in enhancing or suppressing the effect of the previous trial. Specifically this consisted of a binary classification problem aimed at decoding the visual feature (blue or red colour) of the current S1 stimulus during its presentation time window.

To test for differences in the decoding performance across conditions, we applied an ANOVA test with Tukey's multiple comparison test.

All analyses were performed using custom software written in Python; for the decoding algorithm and statistical testing, we used the scikit-learn and Scipy packages.

*Hidden Markov model analysis.* This analysis aimed to study the dynamics of neural activity in an ensemble of simultaneously recorded neurons, using a methodology similar to that adopted in previous works (Benozzo et al. 2021; Marcos et al. 2019; Mazzucato et al. 2019; Ponce-Alvarez et al. 2012). The recording sessions included in the analysis comprised at least four simultaneously recorded neurons, each with a mean trial activity of $\geq 1$ spk/s and at least 25 trials, all of which were preceded by a complete trial. These constraints resulted in the selection of 41 sessions. A detailed description of the fitting and analysis procedure has been previously reported in Benozzo et al. (2021). In that study the procedure was applied to the same dataset but focused on a different time window - from the second stimulus (S2) presentation to the end of the trial. In the current analysis the time window spans two consecutive trials, specifically from the presentation of S2 in the previous trial to the end of S1 presentation in the current trial. The training phase of the model, which involves estimating the transition and emission probability matrices, was performed using the Baum-Welch algorithm. The sequence of states was then decoded from the posterior state probabilities $P(S_k^t | X^t)$ of the state $k$ in presence of data $X$ (the spike train of the ensemble, binned at 5 ms) at time $t$. A state was assigned if its posterior probability was $>0.8$ for at least 50 consecutive ms. Both transition and emission probability matrices required initialization to start the fitting procedure. Specifically the transition matrix was initialized as an identity matrix plus Gaussian noise with zero mean and 0.02 std, and then row-wise normalized to represent probabilities; that is the sum of each row equals one. The emission matrix was initialized by randomly permuting the average firing rate of each neuron, with a Gaussian random term added (zero mean and 0.02 std).

Regarding the number of states, which must be set *a priori*, different models with 2 to $N-1$ states were fitted for each session, where $N$ is the number of neurons in the session. The optimal number of states was chosen to minimize the BIC (Bayesian information criterion) score, calculated as $\text{BIC} = -2\text{LL} + [M(M-1) + MN]\log T$, where LL is the log-likelihood of the model, $M$ is the number of hidden states and $T$ is the number of observations in each session (number of trials × number of bins). The fitting

procedure was repeated 10 times under different initial conditions, with a maximum number of 500 iterations within a 3-fold cross-validation framework; see Maboudi et al. (2018).

After the state sequence was estimated, the presence of a coding state – defined as a state that codes for a specific task variable – was evaluated by testing the mean state occupancy across subsets of trials grouped according to the task variable under analysis. For a given state and time window, mean state occupancy was defined as the average time within the window during which that state was active in a trial (posterior probability >0.8 for a duration ≥50 ms). Their difference was evaluated using a Mann–Whitney *U* rank test with Benjamini–Hochberg false discovery rate correction.

Analysis was performed using custom software written in Python and MATLAB (Statistics and Machine Learning Toolbox, @The MathWorks Inc., Natick, Massachusetts, US).

## Results

### Behavioural results

Two monkeys were trained to discriminate which of two stimuli, S1 and S2, presented sequentially and separated by a first delay (D1) was farther from the centre (Fig. 1*A*).

We tested whether the most recently presented S2 stimulus influenced average trial performance by modulating it in the context of CB. Figure 2*A* shows how the magnitude of the previous S2 affected responses in the current trial, measured as a bias in the probability of selecting S2. We found that as the previous S2 magnitude increased, the S2-ward bias in the current trial decreased (linear fit slope = −1.34, *P* < 0.001; mean number of trials in each (current stimulus, previous S2) pair: 381.9, SD 43.8). That is when the previous S2 had a low magnitude, the S1 stimulus in the current trial tended to be underestimated (Fig. 2*E*, where the colour map indicates the S2-ward bias and the numbers indicate the mean performance variation in each stimulus pair relative to the overall pair performance). This resulted in a higher likelihood of the current response being directed towards the S2 stimulus, and conversely, when the previous S2 had a higher magnitude, there was a greater tendency for the current response to favour the S1 stimulus (Fig. 2*F*). This is in line with how CB affects the performance (see Fig. 2*D* for an explanatory representation of the CB), wherein the first stimulus S1 is contracted towards the mean stimulus distribution (dashed vertical line, drawn in its true position in panel *D*). The contraction of S1 alters the perception of the distance between the two stimuli, in some cases augmenting it, thus making the task easier (green pairs in Fig. 2*D*, bias +), whereas in others reducing it, thus increasing the task difficulty (red pairs in Fig. 2*D*, bias −).

Following this reasoning the behavioural results show that recent trial history influenced the perceived mean of the stimulus. A shift of the stimulus mean towards smaller values as a consequence of a small previous S2 determines a higher probability of underestimating S1 and thus a positive S2-ward bias (see bottom-left inset in Fig. 2*A*; the grey area shows where S1 is underestimated), and the opposite occurs for high recent history (see top-right inset in Fig. 2*A*; the smaller grey area indicates a lower probability of underestimating S1). In Fig. 2*B* and *C* we distinguished trials based on whether there was a colour mismatch or a colour match between the previous S2 and the current S1. On average we observed that the S2-ward bias was more pronounced when there was a colour mismatch (linear fit slope = −1.97, *P* = 0.00309). Conversely the S2-ward bias was significantly reduced when there was a colour match (linear fit slope = −0.70, *P* = 0.0195). A comparison of the S2-ward bias values between the two colour conditions revealed a significant difference (Mann–Whitney *U* rank test: *U* = 2411, *P* = 0.00135). Similarly trials were grouped based on whether the position of the stimulus matched between the current and previous trials. Like colour position is a feature characterizing the stimulus, referring to the side of the screen where it was presented (above or below the central target). However in this case no significant difference was found between the conditions (Mann–Whitney *U* rank test: *U* = 2039, *P* = 0.211). It is important to note that in Fig. 2, we used only stimulus pairs with the smallest relative distance, that is, 8 mm, to avoid other influences on task difficulty.

Four different regression models were fitted to predict the overall behavioural performance (Fig. 2*G*). Specifically two models included only current-trial regressors related to trial difficulty, that is, the stimulus ratio alone (green) and the stimulus ratio combined with the CB (blue). We then extended the model by adding the magnitude of the S2 and S1 stimuli from the previous trial (red and yellow, respectively). Models based only on current-trial features exhibited a significant improvement when the CB regressor was included. Further improvement was observed when the magnitude of the previous S2 was added, allowing $r^2$ to exceed 0.9 (ANOVA test: F(3,796)>1e5, *P* < 0.001, with Tukey's multiple comparison test: blue *vs.* red, *P* = 0.001), whereas including the previous S1 magnitude did not produce a significant effect (red *vs.* yellow, *P* = 0.900). It is worth noting that, although significant, the improvement provided by the previous S2 is relatively small, consistent with the fact that the S2-ward bias induced by the previous S2 is on the order of only a few percentage points (see Fig. 2*A*).

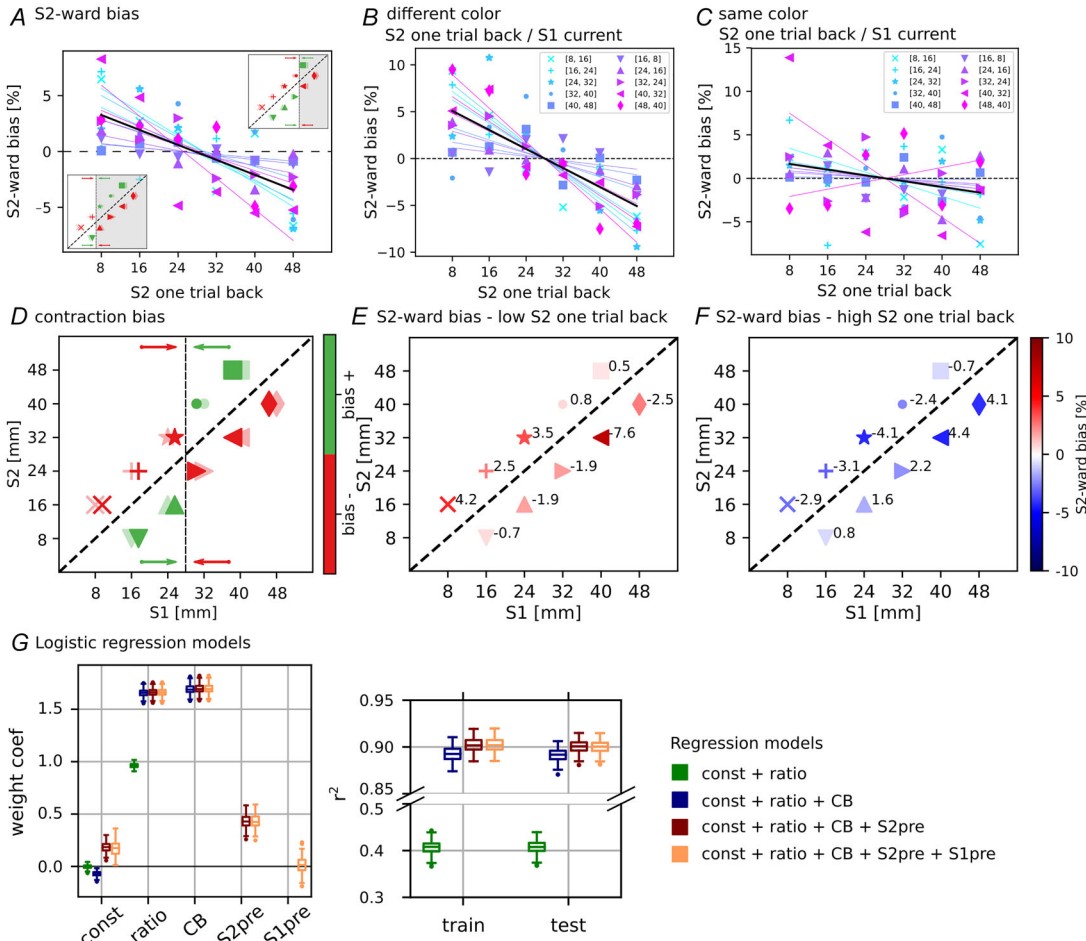

**Figure 2. Behavioural effect of the previous S2 magnitude on the current trial**

*A*, effect of the previous S2 distance on performance. For each stimulus pair, the S2-ward bias relative to the previous occurrence of S2 is plotted (coloured marker), that is the percentage of trials in which S2 was chosen minus the global mean performance of the pair itself. A lower value of the previous S2 tends to elevate the proportion of S2 choices in the present trial and vice versa (linear fit slope = −1.34, *P* < 0.001). Panels *B* and *C* show the same concepts as in panel *A* but with a differentiation based on colour mismatch and colour match between the previous S2 and the current S1. The influence of the previous S2 persists when there is a colour mismatch (linear fit slope = −1.97, *P* = 0.00309), whereas it is reduced when colours are matched (linear fit slope = −0.70, *P* = 0.0195). *D*, assuming an exact stimulus mean perception (black dashed vertical line, 28 mm), the contraction bias (CB) manifests in both positive (green) and negative (red) ways in the current trial (the shape of the marker differentiates stimulus pairs with different relative stimulus distances and therefore different difficulty); previous S2 alters the stimulus mean perception: small values of the previous S2 determine a shift in S1 representation towards lower values that is thus consequently underestimated resulting in a higher percentage of S2 choices (bottom-left inset in panel *A*: grey area includes pairs in which S1 is perceived as smaller, thus favouring the choice of S2, the opposite in the top-right inset: due to a shift towards a higher value of the mean, the grey area becomes smaller; thus the choice of S2 is disfavoured). *E*, alternative representation of the S2-ward bias by selecting only trials with low values of the previous S2, that is, 8 or 16 mm; the colour map indicates the S2-ward bias, and the numbers show the mean performance variation for each stimulus pair relative to the overall pair performance across all previous S2; as in panel *A* the prevalence of red markers indicates a general positive S2-ward bias. *F*, same format as in panel *E*, but here for trials with high values of previous S2, that is, 40 or 48 mm. *G*, four logistic regression models were fitted to predict the probability of choosing S2, across stimulus pairs with a relative distance of 8 mm (10 pairs), and their corresponding previous trial pairs (30 previous pairs each current pair). Two models included only current-trial features: the stimulus ratio (green) and the stimulus ratio plus the CB (blue). In the other two models we added the magnitude of S2pre (red) and additionally the magnitude of S1pre (yellow). Left, distribution of model coefficients for each regressor across models; right, *r²* values of each regressor model computed on the training and testing datasets.

## Neural results: decoding of S1 and influence of the previous S2 stimulus

To investigate the influence of the previous trial in terms of S2-ward bias on neural activity, we focused our analysis on the period of S1 presentation using the prefrontal recordings from the original dataset. First we characterized the neural decoding of S1 itself during its presentation. Figure 3*A* and *B* shows the decoding of S1 starting from 400 ms before its presentation (0 ms) and continuing for the subsequent 1400 ms (comprising 1000 ms of presentation time and an additional 400 ms delay, D1). The stimulus magnitudes ranged from 8 to 48 mm, with 8 mm increments, and were categorized into three broader classes: low (8, 16 mm), medium (24, 32 mm) and high (40, 48 mm). As expected we observed

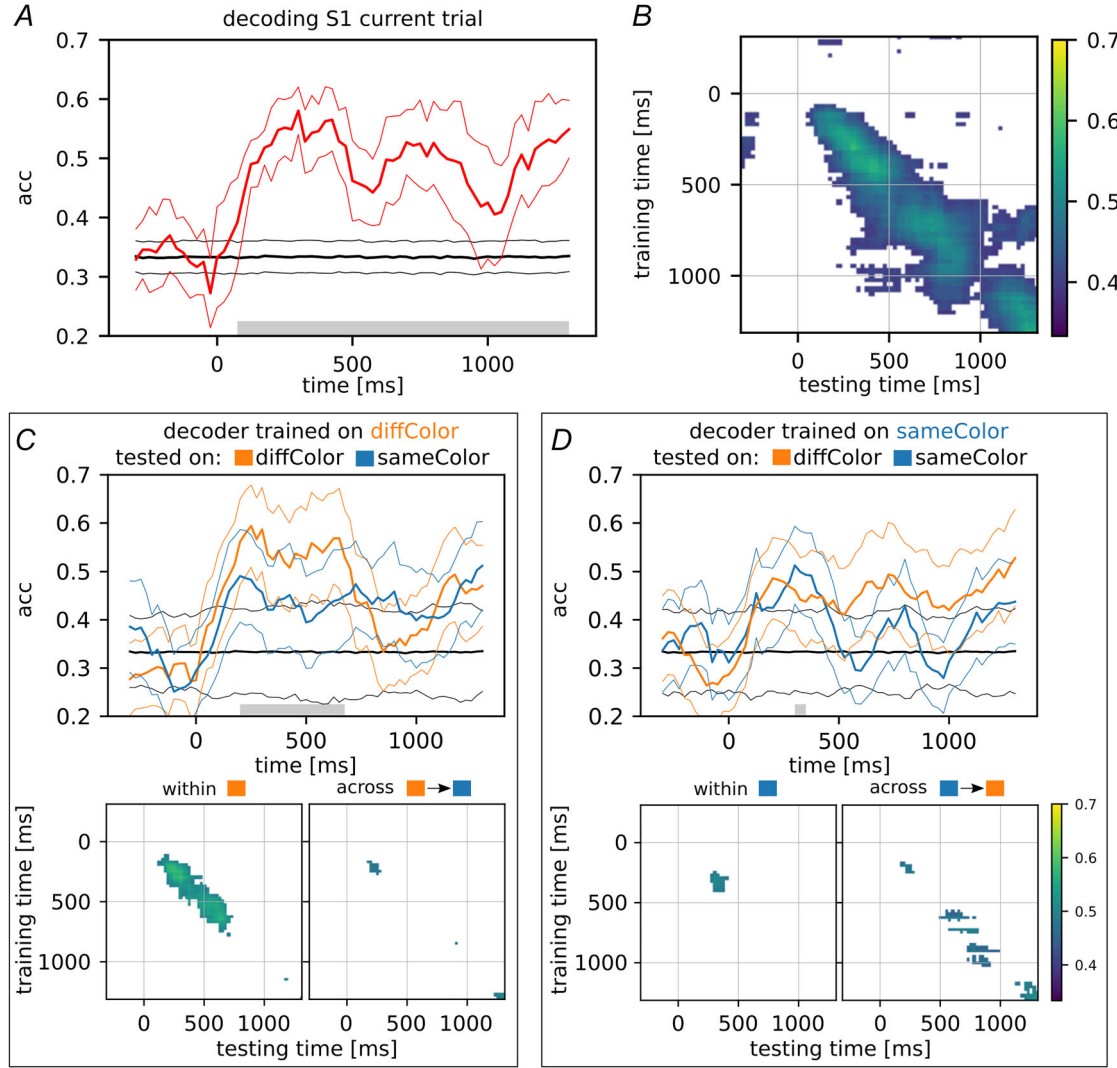

**Figure 3. Decoding analysis of S1 magnitude in the current trial**

*A*, decoding accuracy of stimulus S1 (±SD) grouped into three macro-classes: low (8, 16 mm), medium (24, 32 mm), and high (40, 48 mm). Accuracy is shown from 400 ms before S1 appearance (0 ms) until 400 ms after the end of its presentation period (which lasted for 1000 ms; the final 400 ms is the shortest delay D1 in common with all trials) (bin size: 200 ms, step size: 25 ms, 100 repetitions with random train/test split). The grey bar over the time axis indicates a significant difference from chance (random permutation test, 1000 iterations). *B*, same decoding performance evaluated across training and testing time windows (the diagonal corresponds to the curve in panel *A*). Panels *C* and *D* illustrate the same concept as shown in panels *A* and *B* but with a distinction based on whether there is a colour mismatch (diffColor) or colour match (sameColor) between the previous S2 and the current S1. Decoding accuracies for both within- and across-condition analyses are displayed, accompanied by the corresponding training/testing time maps (bottom section). White shading in the maps indicates no significant deviation from chance levels. To reproduce these panels and compute the related *P*-values, refer to the 'Data Availability Statement'.

significant decoding accuracy for S1 both throughout the entire presentation period and during D1 (random permutation test, $n = 1000$; in Fig. 3*A* and *B* significant results are shown by the grey bar over the time axis, and by the coloured area on the map, with white indicating non-significance). In particular panel *B* shows how the decoding performance changed dynamically when evaluated across all pairs of training and testing time windows (diagonal elements: same training and testing time windows, as shown in panel *A*; off-diagonal elements: different training and testing time windows). This dynamic coding indicates that a different coding scheme is used by the prefrontal population of neurons for coding S1 over time.

Figure 3*C* and *D*, similar to the previous panels, shows the decoding of S1 categorizing the trials into those with a colour mismatch (diffColor) and those with a colour match (sameColor) between the previous S2 and the current S1. This categorization had a significant impact on the decoding performance, leading to a significant decrease, likely caused by the reduction to half of the number of trials and the disentanglement of the colour matching feature. However it is worth noting that decoding accuracy remains higher in the diffColor condition, further reinforcing its role, as also evidenced in the behavioural data shown in Fig. 2*B* and *C*.

To assess the impact of the previous trial on the decoding accuracy of S1, we conditioned its classification on the S2 distance of the preceding trial. We maintained the same macro-level groups (low, medium, and high magnitude) for the previous S2 as well and clustered each trial accordingly. Due to the randomization of stimuli across trials, we anticipated a balanced distribution of the S1 classes (low, medium, shown and high) within each S2 one-trial back class. This approach enabled us to characterize the neural decoding of S1 for each category of S2 one-trial back. Figure 4*A* and *C* shows the mean decoding accuracy of S1 within and across the previous S2 conditions, whereas Fig. 4*B* and *D* shows the variation in the marginal probability of each S1 class across decoding conditions. More specifically we examined how the marginal probability of each S1 class varied when decoders trained on the two extreme cases, low and high S2 one-trial back, were applied to decode S1 on trials with a different history. Figure 4*A* and *B* and *C* and *D* refers to the first (early S1) and second (late S1) half-periods of S1 presentation, respectively.

In general we observed that the highest decoding accuracy was achieved when the training and testing trials used to decode S1 had similar previous S2 values, indicating an advantage of decoding S1 within the same S2 one-trial back condition. These accuracy values closely align with the results in Fig. 3*A*, where the previous trial's S2 was not taken into account. However the ability to decode S1 significantly diminished in cases where there

was a discordant S2 one-trial back condition. Particularly the reduction in accuracy corresponded to the degree of disparity with the S2 values from one-trial back: the greater the discrepancy, the more pronounced the decline in S1 accuracy. This pattern is shown in Fig. 4*A* and *C*, right panels. The column labelled 'same S2 one back' pertains to within-condition decoding, whereas the other two columns represent across-condition decoding, which was further divided into 'low $\Delta$S2 one back' and 'high $\Delta$S2 one back'. 'Low $\Delta$S2 one back' indicates low discrepancy between S2 one-trial back in training and testing trials, for example, *low* S2 one back in training and *medium* in testing, whereas 'high $\Delta$S2 one back' indicates high discrepancy, for example, *low* S2 one back in training and *high* in testing. This result was particularly prominent in the early S1 period (as observed in panel *A*, ANOVA test: $F(2,117) = 21.593$, $P < 0.001$) and gradually diminished as S1 progressed into its later stages (as observed in panel *C*, ANOVA test: $F(2,112) = 4.675$, $P = 0.0112$), eventually fading during the D1 period.

In summary these findings indicate that when a decoder, initially trained to distinguish S1 using trials with a specific S2 history, is subsequently tested in a trial with a different S2 history, its performance declines. This result suggests that the S2 stimulus from one-trial back influenced the perception of S1 in the current trial, leading to a disruption in the decoder's ability to accurately classify S1. To gain further insight into this phenomenon, we conducted the analysis shown in panels *B* and *D*, which illustrates the variations in marginal probabilities of predicting the S1 class in relation to the S2 one-trial back magnitudes used to train the decoder. Particularly given a decoder trained to discriminate S1 in trials with low S2 one-trial back, we computed the marginal probabilities of assigning a low, medium or high label to trials with the same previous S2 values. These probabilities serve as a reference, as they are not influenced by history bias (training and testing trials share the same S2 one-trial back category). We then applied the same decoder to trials with medium and high S2 one-trial back, and computed how the marginal probabilities deviated from the reference case. In this way we quantified whether the decoder's predictions were biased when applied to trials with a different previous S2 than the one it was trained on. Indeed, it emerged that the decoder's predictions were biased towards its own training previous S2 in the early S1 period (ANOVA test: $F(2,57) = 19.319$, $P < 4e-7$, low *vs.* high: $P < 0.001$; in late S1, ANOVA test: $F(2,57) = 8.064$, $P < 0.001$, low *vs.* high: $P = 0.107$). Similarly when trained on high previous S2 values and tested on low and medium previous S2 values, the decoder's predictions showed a comparable bias towards its own training previous S2, in both the early and late S1 periods (ANOVA test: $F(2,57) = 22.342$, $P < 7e-8$, low *vs.* high: $P < 0.001$; ANOVA test: $F(2,57) = 37.158$, $P < 1e-11$, low *vs.* high:

$P < 0.001$, respectively). It is important to emphasize that this analysis is not intended to reproduce the neural mechanisms underlying the history bias in S1 encoding. Rather it shows that a linear decoder exhibited a similar bias that could account for the observed behavioural results.

In the lower part of Fig. 4, panel *E* shows the results of the same analysis shown in the previous panels, conducted

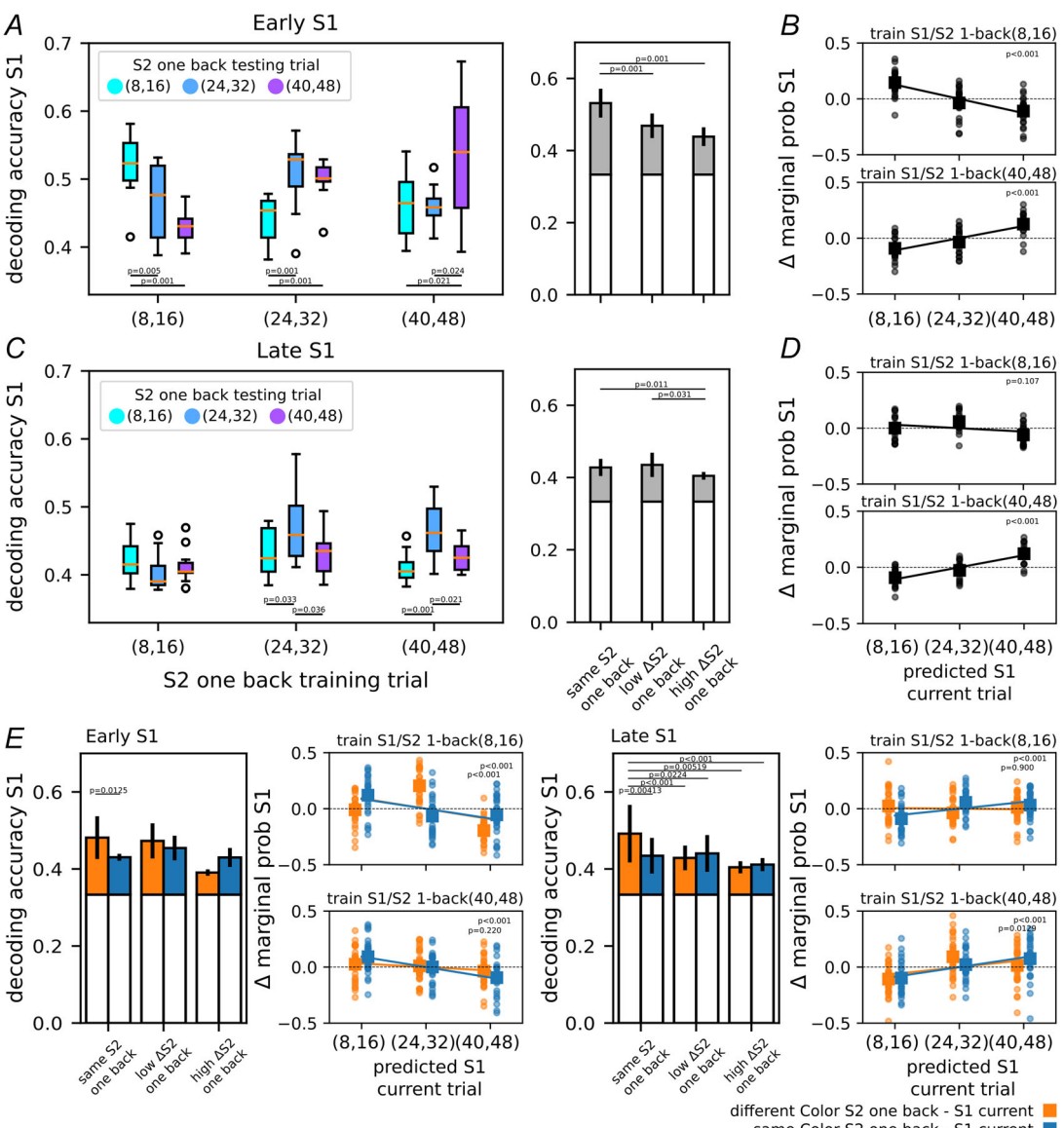

**Figure 4. Impact of the previous trial on S1 decoding accuracy**

*A*, decoding accuracy of stimulus S1 conditioned on S2 of the previous trial; the boxplots show how the accuracy changes in the first half of S1 presentation (early S1) when training and testing trials share (or do not share) their S2 in one-trial back (left side). On the right side of the panel, the same results as in the boxplots but grouped as follows: decoding of S1 within condition (same previous S2 trial) and across conditions (low ΔS2: small gap between S2 one-trial back in training and test sets, e.g. (8,16) *vs.* (24,32), high ΔS2: large gap between S2 one-trial back in training and test sets, e.g. (8,16) *vs.* (40,48)). *B*, how the marginal probability of each S1 class varied when decoders trained on the two extreme cases, S2 one-trial back low and high (top and bottom parts, respectively), were applied to decode S1 on trials with a different history (the reported significance refers to the difference between the two furthest classes: S1 low *vs.* S1 high). Similarly panels *C* and *D* focus on the last half of S1 presentation (late S1). In panel *E*, following a structure analogous to that of panels *A* and *C*, but with trials divided by colour mismatch (orange bars) or match (blue bars) between the previous S2 and the current S1. ANOVA test with Tukey's multiple comparison test.

separately for the diffColor and sameColor conditions. Regarding the reduction in S1 decoding accuracy as the discrepancy from the previous S2 increased, this pattern was preserved in the diffColor condition, though it was more pronounced during the late S1 time window (late S1). A two-way ANOVA was performed with colour condition and discrepancy degree as factors, and the interaction term was significant in both time windows ($F_{(2,139)} = 3.087$, $P = 0.0407$, and $F_{(2,133)} = 7.456$, $P < 0.001$, respectively). No significant differences were observed across discrepancy levels in the sameColor condition. Concerning the variation in the marginal probabilities for each predicted S1 class, during the early S1 window, the decoder bias towards its training S2 one-trial back values was observed in both colour conditions in the case of low previous S2 ($P < 0.001$). However for decoders trained on high previous S2 values, the diffColor condition showed no bias ($P = 0.220$), whereas in the sameColor condition, the bias unexpectedly shifted towards the S2 values of the testing trials (rather than those of the training trials) ($P < 0.001$). A symmetric pattern emerged during the late S1 window: the bias was consistent with expectations for decoders trained on high previous S2 values in both colour conditions ($P = 0.0129$ and $P < 0.001$), but in the sameColor condition, decoders trained on low previous S2 values exhibited a reversed bias ($P < 0.001$), whereas no bias was observed in the diffColor condition ($P = 0.900$). Statistical significance values reported here and in the corresponding panel insets refer to comparisons between the two extreme S1 classes, that is, (8,16) *vs.* (40,48).

## Neural results: hidden Markov model analysis and population decoding of S1 colour

The preceding section provided evidence of interference from the previous S2 on the decoding of the current S1, indicating that stimulus features can influence this process. This suggests the possibility that both variables, namely S2 from one-trial back and S1 colour, might be decoded during the presentation of S1, as they both contributed to determining the ability to decode the magnitude of S1.

Regarding the colour feature we studied each recording session separately by applying a hidden Markov model (HMM) analysis and also using a population decoding approach similar to what was described in the previous section. Using the session-level HMM analysis, we examined trial-wise state sequences to identify coding states for S1 colour (blue *vs.* red), separately for diffColor and sameColor trials. This analysis yielded the following results: 6 out of 41 sessions ($P = 0.012$, binomial test) contained a state whose mean occupancy differed significantly between red and blue stimuli in the diffColor

condition, whereas 5 out of 41 sessions ($P = 0.037$, binomial test) showed such a state in the sameColor condition. Interestingly these coding states were largely condition specific: state encoding colour in one condition did not encode colour in the other (with the exception of a single session in which the same state encoded colour in both conditions). This pattern suggests that S1 colour encoding depends on the previous-trial context. Figure 5*A* shows two examples of sessions with coding states for the stimulus colour feature. The panel reports the mean state occupancy for each state, separately for the two conditions and the two colours, computed both during the stimulus presentation window (first row) and during the pre-stimulus delay (second row). Subplots with reduced transparency refer to states that are not coding states (particularly none of them are coding states during the pre-stimulus delay). The last row of the panel shows the firing rate profile for each state. A more comprehensive visualization of state sequences is shown in Fig. 5*B*, where the HMM state sequence for each coding state is reported at the single-trial level, spanning from the appearance of S2 in the previous trial to the end of S1 presentation in the current trial.

Regarding the population decoding analysis we also assessed the ability to classify each trial based on the colour of S1. Figure 6*A* shows the accuracy curve, revealing a significant peak during the first half of the presentation window. Figure 6*B* shows the mean accuracy values for 'early S1' and 'late S1' (red boxplots). Additionally the panel includes the mean accuracy values for these two periods in both the diffColor and sameColor conditions, illustrated in orange and blue boxplots, respectively. In line with our previous findings, the sameColor condition was associated with reduced discriminability of the stimulus features ($P < 0.001$ and $P = 0.00347$, early and late S1 periods, respectively). Figure 6*C* shows the training/testing time map. This map illustrates the scenario without trial separation, with contour lines delineating the expansion of the map in the context of colour mismatch (diffColor), whereas no significant accuracies were observed when the colours matched (sameColor). Both analyses provide evidence of the importance of the visual features in the neural data, specifically showing a relationship with the colour feature of the previous stimulus.

Focusing on S2 of one-trial back, despite its irrelevance to task performance but due to its impact on S1 decoding, we explored whether there was still a trace of it in the current trial. Figure 6*D* shows that after an initial encoding of the magnitude of the previous S2 (before the presentation of S1, 0 ms), it was no longer decoded until the second half of S1 presentation (late S1). During this latter period it reached its highest accuracy, nearly 0.6, whereas it was at chance level in the early S1 period. Interestingly this peak of accuracy for decoding the previous S2

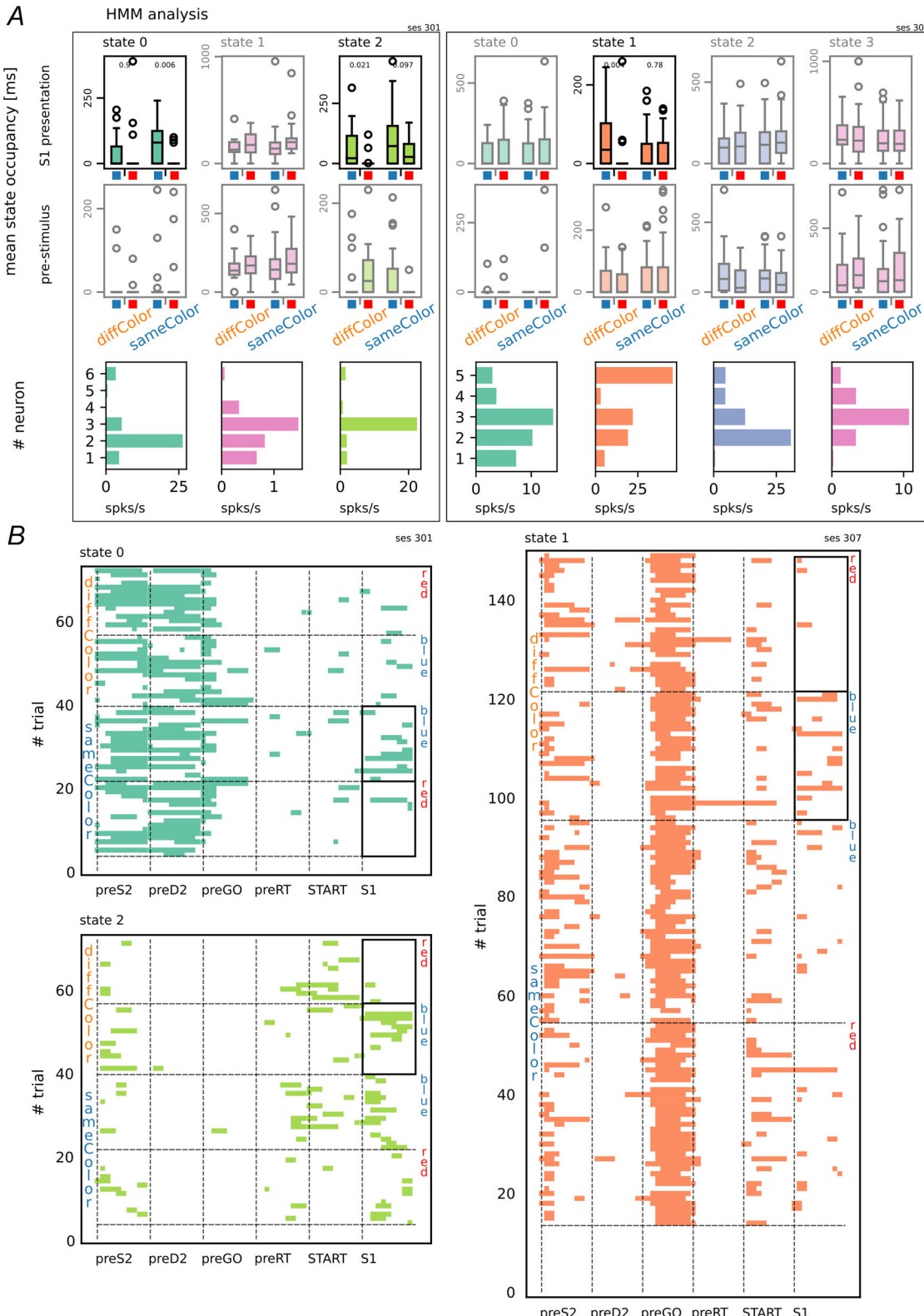

**Figure 5. Hidden Markov model analysis**

*A*, examples of sessions with significant HMM (hidden Markov model) coding states for the first stimulus colour feature. In each session's panel the mean state occupancy is shown for each state, condition (diffColor *vs*. sameColor) and stimulus colour (blue *vs*. red). The first row refers to the time window of S1 presentation, whereas the second row covers the pre-stimulus delay interval. States with no significant mean occupancy between colours

are displayed with reduced transparency (Mann–Whitney *U* rank test with Benjamini–Hochberg false discovery rate correction). The last row shows the firing rate profile of each state. Panel *B* extends panel *A* by presenting the HMM state sequences for each coding state at the single-trial level, spanning from the appearance of S2 in the previous trial (preS2) to the end of S1 presentation in the current trial. Trials are grouped by the diffColor or sameColor condition and, within each condition, by the stimulus visual feature (blue or red). The presence of coding states is represented by black bold rectangles, which enclose subsets of trials where the mean state occupancy significantly differs between different stimulus visual features.

appeared later than the time window in which the effect of S2 history on the decoding of S1 was observed. This raises the possibility that a silent trace of the previous S2 distance might influence the coding of S1 without being explicitly represented, as we will propose in the 'Discussion'. In Fig. 6*E* the decoding performance is visualized across all pairs of training and testing time windows. Two distinct clusters are evident along the diagonal. The first, with lower intensity, is located before 0 ms, corresponding to the initial peak observed in Fig. 6*D*. The second cluster,

the most prominent one, is centred in the late S1 period, aligning with the highest peak in Fig. 6*D*. The white area in the map indicates no significant decoding during those time intervals (random permutation test $n = 1000$). This panel suggests that there was no discernible relationship between the coding of S2 before S1 and the late S1 period, as no significant decoding was observed across these two periods. This lack of significance hints at a potential change in the population coding between these time intervals, indicating that the processing of information

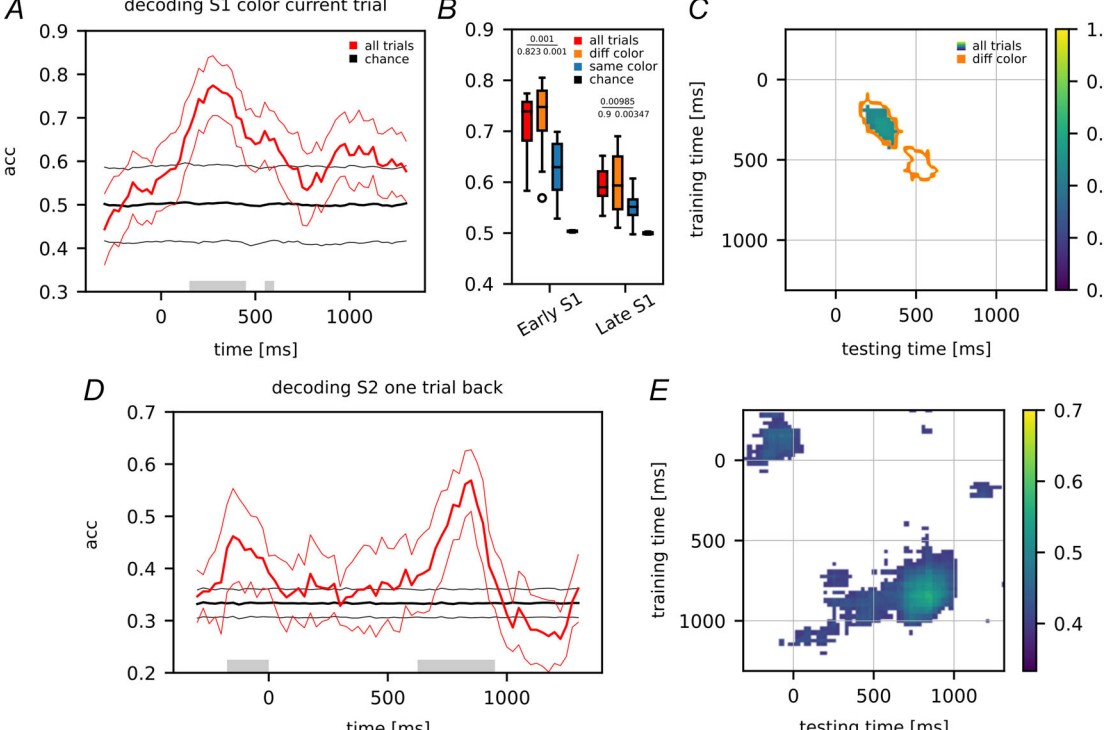

**Figure 6. Decoding of current stimulus color and reactivation of the previous S2 neural representation**
*A*, Decoding accuracy of the colour (blue *vs.* red) of stimulus S1 (±SD). Accuracy is shown from 400 ms before S1 appearance (0 ms) until 400 ms after the end of its presentation period (which lasted for 1000 ms; the final 400 ms is the shortest delay D1 in common with all trials) (bin size: 200 ms, step size: 25 ms; 100 repetitions with random train/test split). The grey bar over the time axis indicates a significant difference from chance (random permutation test, 1000 iterations). *B*, boxplots compare decoding accuracies when S1 colour mismatches (diffColor, orange boxplot) or matches (sameColor, blue boxplot) with the colour of the previous S2; the red boxplot reports the global accuracy without separating trials on colour conditions. *C*, decoding performances evaluated across different training and testing time windows: the map illustrates the scenario without trial separation, the contour lines delineate the expansion of the map in the context of colour mismatch, whereas no significant accuracies were observed when colours matched. *D*, decoding accuracy of the previous S2 (±SD) during S1 presentation (current trial). The previous S2 stimulus distances are grouped in three macro-classes: low, medium, and high distances. Same approach as in panel *A*. *E*, the decoding performance evaluated across different training and testing time windows. To reproduce these panels and compute the related *P*-values, refer to the 'Data Availability Statement'.

changed at the population level across different neurons as the trial progressed from the early to late S1 period. As we did for the colour matching feature in the study of S1 magnitude decoding, we also examined the effect of colour match/mismatch when applied to S2 one-trial back. However no significant results were observed in either the match or mismatch conditions; the related curves indicated a reduction towards chance level and a shift of the accuracy peak to the early S1 presentation window, respectively.

## Discussion

We studied the influence of the most recently presented stimulus on the performance of monkeys in a distance discrimination task to investigate its neural bases in the prefrontal cortex and whether we could find signatures already in the stimulus coding of this influence. Specifically we investigated whether the most recent stimulus could modulate the CB effect and interfere with the coding of the stimulus to discriminate. Consistent with the findings of Benozzo et al. (2023) regarding the average distribution of stimuli, we also found short-term effects of the previous trial with the previous S2 significantly affecting performance, contributing to the modulation of the CB effect. Specifically we identified an attractive effect of the previous S2 magnitude, which appeared as a shift in the representation of the S1 magnitude towards the previous S2. This was also supported by a logistic regression model of the average single-pair performance, which achieved a higher predictive performance when the previous S2 was included together with difficulty-based regressors such as the CB and the stimulus ratio. As a possible neural substrate we found that the magnitude of the previous S2 affected the decoding of S1. S1 was decoded with the highest accuracy when the decoder was trained and tested with S2 of the same magnitude, whereas a mismatch in magnitudes altered its decoding, as shown by the reduction in the decoding. The greater the mismatch, the higher the reduction in accuracy. Moreover when the decoder was applied to trials with a different previous S2 value than the one it was trained on, it biased the predicted S1 class towards the previous S2 value used during training. As discussed in a later section, we also found that the effect of the previous trial depended on whether the visual features of S1 differed from those of the previously presented S2.

### History bias

Although our analysis does not provide a mechanistic explanation, the effect of the previous S2 on the representation of S1 suggests that biases generated from the previous trial can influence the sensory representation of the current stimulus, and not only the decision process, as already shown in our previous work (Benozzo et al., 2023). In that study we demonstrated that the decision process is affected by the mean distribution of stimulus magnitudes, but we were unable to determine whether the representation of the first stimulus's magnitude was also influenced by CB. This limitation arose because comparing the effect of different mean distributions on the encoding of the first stimulus would have been necessary. In the present study we addressed this limitation by investigating how the previous trial affects the encoding of the first stimulus when the second stimulus has varying magnitudes. Our findings reveal that biases from previous trials can indeed alter the decoding of the stimulus to be discriminated. This suggests that the decision-related activity observed after both stimuli are presented may be driven, at least in part, by a biased representation of the first stimulus's magnitude at the time it is presented. It is important to note that, because our conclusions are based on the properties of a decoder trained to discriminate task variables under different history conditions, this approach should not be interpreted as a neural model replicating the mechanisms underlying the history bias in S1 encoding. Rather it shows that a linear decoder could replicate a similar bias when applied to trials with significantly different history conditions from those it was trained on.

Our results on the influence of previous trials resemble the findings of Histed et al. (2009) on the effect of history on the current-trial coding activity. They showed that the previous outcome information was maintained across trials and that task variables' performance and coding, such as response direction, were encoded stronger after correct than error trials, possibly favouring learning. Our result is also in line with the behavioural results of Papadimitriou et al. (2015) in monkeys, who found that memory in the current trial shows a bias towards the location of the memorandum from the previous trial (proactive interference). The effect of the history bias on performance was also reported in rats by Akrami et al. (2018), where the activity of the posterior parietal cortex revealed a causal role in modulating the impact of recent trials on performance. Similarly Barbosa et al. (2020) found a stronger effect of previous trials linked to a reactivation of activity in the monkey and human prefrontal cortex during the initial phase of the trial, which was previously silent during the intertrial interval. More specifically regarding the role of the previous S2 in affecting the decoding of the current S1 (see Fig. 4), Boboeva et al. (2024) recently developed a computational model that aims to reproduce the interplay between the parietal cortex and the working memory network. Their model showed a direct relationship between the contraction of S1 and its difference from the previous trial's stimulus S2. This indeed strongly resonates with

our findings, where S1 decoding is maximized when the decoder was trained and tested on trials with similar previous S2 magnitude, and proportionally decreased with increasing their discrepancy.

### Reactivation of neural traces from the previous trial

We searched for a trace of the past S2 magnitude during the S1 presentation that could account for the previous trial effects. We found that the previous S2 modulated the current-trial activity in the pre-stimulus period but decreased to chance levels after the presentation of S1. Its modulation reappeared again in the latest S1 period, as if the memory of the previous S2 resurfaced only after a period of no explicit representation. Particularly this finding refers specifically to the encoding modality of the previous S2 in the recording area under analysis; it does not rule out the possibility that other brain areas may encode the same information using a different modality. The first study showing an interruption of coding in time, conceptualized as a silent activity, is a study by Watanabe and Funahashi (2014). Recording from the prefrontal cortex, using a dual task paradigm, they found that the memory activity associated with a location to be held in memory disappeared temporarily during an intervening attentional task, only to emerge again later when the concurrent task was completed. The reactivation of the representation of the previous stimulus in the pre-stimulus interval, right before the appearance of the new stimulus, also resembles the effect observed by Barbosa et al. (2020) and represents a confirmation of this phenomenon. According to Barbosa and colleagues a break in the continuity of coding should be interpreted not as a loss of information in the network but as a sign of a switch to a different modality of information maintenance, termed 'activity silent'. Being in an 'activity-silent' state has been proposed to support working memory (Stokes, 2015). The mechanism underlying 'activity-silent' states is not yet clear; one proposed mechanism is that information is maintained in memory through a temporary modification of the synaptic weights (Barak and Tsodyks, 2014; Kaminsky and Rutishauser, 2020; Mongillo et al., 2008). 'Activity-silent' network dynamics would represent an alternative mechanism to persistent and dynamic coding (Ceccarelli et al., 2023; Mendoza-Halliday and Martinez-Trujillo, 2017; Meyers et al., 2008; Meyers et al., 2012; Meyers, 2018) to keep information in working memory. In addition to neurophysiology some evidence for activity-silent forms of memory maintenance comes from neuroimaging studies, for example, using retro-cue paradigms (Rose et al., 2016). In this line Ranieri et al. (2022) reported the activation of an activity-silent trace related to the previous response by applying decoding analysis on EEG recording during a spatial frequency discrimination task. Particularly the peak in decoding accuracy occurred 700 ms after the presentation of the current stimulus, similar to the time window in which we observed the reactivation of the previous S2. However imaging studies have the limitation that they cannot determine whether the absence of decoding capability reflects a complete loss of coding activity at the single-cell level or just a reduction in the number of cells involved (Kaminsky et al., 2020). Although the timing of the reactivation of previous trial information in our study aligns with the findings of Ranieri et al. (2022), it differs from Barbosa et al. (2020), where reactivation occurred right before stimulus presentation, together with the ramping up of neural activity. Why information is kept in an 'activity-silent' state in some cases but not in others and why its timing varies remain unclear and might depend specifically on the task structure and task requirement. In our study the reactivation of the previous S2 representation occurred primarily during the late S1 period, whereas the initial phase of S1 presentation was most influenced by the previous S2 value, at least when considering all trials regardless of the visual features of the previous stimulus. It is not clear why an explicit representation of S2 was lost during the initial representation of S1. One possible interpretation is that the shift to an 'activity-silent' representation resulted from the interaction between current and past neural representations, where neural resources are prioritized for processing the current stimulus over the past one because it is more relevant. It is possible that the resources allocated in our task for other computations, such as the representation of the stimulus features of the first stimulus, dominated the competition for relevance over the information about the previous stimulus.

### Effect of visual stimulus features

We also observed an unexpected effect: the influence of the matching of visual stimulus features across trials on the emergence of the bias. The fact that visual stimulus features influenced the serial bias on the following trial, whereas the spatial feature (i.e. whether the stimulus was shown above or below the central target) did not, echoes the findings of Bae and Luck (2020). In their study serial dependence selectively affected the feature that had been reported in the previous trial, which in our task always corresponds to the visual feature. Why is the behavioural bias from the previous trial attenuated when the current stimulus shares the same visual features as the previous S2? One tentative explanation is that the reduction in the first stimulus feature coding depends on an adaptation effect (Barron et al., 2016; Gross et al., 1967; Henson, 2003; Karlaftis et al., 2021). This effect has been shown also in monkeys, not only in the IT cortex (Gross et al., 1967; Miller et al., 1991) but also in

the prefrontal cortex by Miller et al. (1996) and Rainer et al. (1999). Karlaftis and colleagues (2021) showed in a functional magnetic resonance imaging (fMRI) study in humans a decrease in activity in the PFdl after the repeated presentation of sensory stimuli. We hypothesize that the repetition of the same visual features affects other computations. First it would reduce the response to the visual features of the first stimulus presentation. Second the adaptation could simultaneously influence concurrent representations, such as the magnitude of S1 (Fig. 3*D*) and the impact of information from the previous S2 on S1 coding (Fig. 4*E*). This hypothesis is further supported by the computational model of the working memory network proposed by Boboeva et al. (2024), which includes an adaptation term that adjusts the persistence of self-sustained activity related to past stimuli. The stronger the adaptation term, the faster the extinguishing process of the previous trial's activity. Currently this remains speculative, and future studies are needed to understand this effect and to test whether it can be generalized to humans, other species, paradigms and sensory modalities.

Although the CB effect and to a lesser degree the effect of the last stimulus could explain a great part of the performance, other factors could have affected the performance. One possibility is the exploration of alternative strategies (Bussey et al., 2001; Fascianelli et al., 2024; Genovesio et al., 2005), but in our task, no alternative strategy could be used to improve performance, although it is possible that the monkeys explored certain strategies in some sessions. At times monkeys may have become distracted and failed to attend to the stimulus distance; in such cases their decisions would resemble those made in a free-choice task. In a previous study (Marcos and Genovesio, 2016) we found that a small imbalance in the activity of directional cells tuned to different target positions could bias the subsequent choice under free-choice conditions.

## Conclusion

We found two main results. First we observed a short-term effect of the previous trial on performance, specifically an attractive influence of the previous stimulus S2 on the current first stimulus S1; that is S1 tended to be perceived as shifted towards the S2 presented in the preceding trial. At the neural level we detected a signature of S2 influence already during S1 processing, indicating that the previous stimulus can affect the discrimination process starting from the neural activity associated with S1, that is at early stages of stimulus processing rather than only at the decision stage. Second we found that information about the previous S2 could not be decoded during the initial phase of S1 presentation, suggesting that it remained in

what has been described as an 'activity-silent' state. We interpret this as reflecting a transient prioritization of the processing of the current stimulus over the past one due to its higher behavioural relevance. However further modelling studies are needed to support this hypothesis.

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

## Additional information

### Data availability statement

Script container to reproduce decoding accuracy curves and training/testing time maps with related *P*-values: https://github.com/danilobenozzo/paper_historyBias.

Source data are available from the authors upon reasonable request.

### Competing interests

The authors declare no competing interests.

### Author contributions

D.B.: conception or design of the work; acquisition, analysis or interpretation of data for the work; drafting the work or revising it critically for important intellectual content; approving the final version of the manuscript; agreement to be accountable for all aspects of the work. L.F.: acquisition, analysis or interpretation of data for the work; drafting the work or revising it critically for important intellectual content; approving the final version of the manuscript; agreement to be accountable for all aspects of the work. F.C.: acquisition, analysis or interpretation of data for the work; drafting the work or revising it critically for important intellectual content; approving the final version of the manuscript; agreement to be accountable for all aspects of the work. A.G.: conception or design of the work; acquisition,

analysis or interpretation of data for the work; drafting the work or revising it critically for important intellectual content; approving the final version of the manuscript; agreement to be accountable for all aspects of the work.

## Funding

This work was supported by Ministero dell'Università e della Ricerca (grant PRIN 2017: 2017KZNZLN_004) and by Sapienza University (Ateneo 2023: RM123188F6C0E21E). D.B. was supported by NEXTGENERATIONEU (NGEU) and funded by the Ministry of University and Research (MUR), National Recovery and Resilience Plan (NRRP), project MNESYS (PE0000006) – a multiscale integrated approach to the study of the nervous system in health and disease (DN. 1553 11.10.2022).

## Acknowledgements

We thank Dr Satoshi Tsujimoto and Dr Steven P. Wise for their contribution to the experimental and data acquisition aspects and Dr Andrew Mitz for help with engineering.

Open access publishing facilitated by Universita degli Studi del Piemonte Orientale Amedeo Avogadro, as part of the Wiley - CRUI-CARE agreement.

## Keywords

contraction bias, distance discrimination task, history bias, monkeys, prefrontal cortex, silent coding

## Supporting information

Additional supporting information can be found online in the Supporting Information section at the end of the HTML view of the article. Supporting information files available:

**Peer Review History**

