## [Peer Review History · The Journal of Physiology]

History bias and its perturbation of the stimulus representation in the macaque prefrontal cortex.

Danilo Benozzo, Lorenzo Ferrucci, Francesco Ceccarelli, and Aldo Genovesio
DOI: 10.1113/JP288070

Corresponding author(s): Aldo Genovesio (aldo.genovesio@uniupo.it)

The following individual(s) involved in review of this submission have agreed to reveal their identity: Ken D O'Halloran (Referee #1)

Review Timeline:

Submission Date:	11-Nov-2024
Editorial Decision:	06-Feb-2025
Revision Received:	17-Jun-2025
Editorial Decision:	30-Jul-2025
Revision Received:	15-Nov-2025
Editorial Decision:	06-Jan-2026
Revision Received:	27-Jan-2026
Accepted:	03-Feb-2026

Senior Editor: Richard Carson

Reviewing Editor: Ricci Hannah

Transaction Report:

Dear Dr Genovesio,

Re: JP-RP-2024-288070 "History bias and its perturbation of the stimulus representation in the macaque prefrontal cortex." by Danilo Benozzo, Lorenzo Ferrucci, Francesco Ceccarelli, and Aldo Genovesio

Thank you for submitting your manuscript to The Journal of Physiology. It has been assessed by a Reviewing Editor and by 3 expert referees and we are pleased to tell you that it is potentially acceptable for publication following satisfactory major revision.

REVISION CHECKLIST:

Please upload two versions of your manuscript text: one with all relevant changes highlighted and one clean version with no

changes tracked. The manuscript file should include all tables and figure legends, but each figure/graph should be uploaded as separate, high-resolution files.

We look forward to receiving your revised submission.

Yours sincerely,

Richard Carson
Senior Editor
The Journal of Physiology

REQUIRED ITEMS

- Author photo and profile. First or joint first authors are asked to provide a short biography (no more than 100 words for one author or 150 words in total for joint first authors) and a portrait photograph. These should be uploaded and clearly labelled together in a Word document with the revised version of the manuscript. See Information for Authors for further details.

- You must start the Methods section with a paragraph headed Ethical approval (https://jp.msubmit.net/cgi-bin/main.plex?form_type=display_requirements#methods).

Research must comply with The Journal's policies regarding animal experiments (<https://physoc.onlinelibrary.wiley.com/hub/animal-experiments>) and adherence to these policies must be stated in the manuscript.

Authors should confirm in their Methods section that their experiments were carried out according to the guidelines laid down by their institution's animal welfare committee, including an ethics approval reference number. The Methods section must contain a statement about access to food, water and housing, details of the anaesthetic regime: anaesthetic used, dose and route of administration, and method of killing the experimental animals.

- Your manuscript must include a complete Additional Information section, including competing interests; funding; author contributions and acknowledgements.

- Please upload separate high-quality figure files via the submission form.

- Your paper contains Supporting Information of a type that we no longer publish, including supplementary tables and figures. Any information essential to an understanding of the paper must be included as part of the main manuscript and figures. The only Supporting Information that we publish are video and audio, 3D structures, program codes and large data

files. Your revised paper will be returned to you if it does not adhere to our Supporting Information Guidelines.

- Papers must comply with the Statistics Policy: https://jp.msubmit.net/cgi-bin/main.plex?form_type=display_requirements#statistics.

In summary:

- If n {less than or equal to} 30, all data points must be plotted in the figure in a way that reveals their range and distribution. A bar graph with data points overlaid, a box and whisker plot or a violin plot (preferably with data points included) are acceptable formats.

- If $n > 30$, then the entire raw dataset must be made available either as supporting information, or hosted on a not-for-profit repository, e.g. FigShare, with access details provided in the manuscript.

- 'n' clearly defined (e.g. x cells from y slices in z animals) in the Methods. Authors should be mindful of pseudoreplication.

- All relevant 'n' values must be clearly stated in the main text, figures and tables.

- The most appropriate summary statistic (e.g. mean or median and standard deviation) must be used. Standard Error of the Mean (SEM) alone is not permitted.

- Exact p values must be stated. Authors must not use 'greater than' or 'less than'. Exact p values must be stated to three significant figures even when 'no statistical significance' is claimed.

- Please include an Abstract Figure file, as well as the Figure Legend text within the main article file. The Abstract Figure is a piece of artwork designed to give readers an immediate understanding of the research and should summarise the main conclusions. If possible, the image should be easily 'readable' from left to right or top to bottom. It should show the physiological relevance of the manuscript so readers can assess the importance and content of its findings. Abstract Figures should not merely recapitulate other figures in the manuscript. Please try to keep the diagram as simple as possible and without superfluous information that may distract from the main conclusion(s). Abstract Figures must be provided by authors no later than the revised manuscript stage and should be uploaded as a separate file during online submission labelled as File Type 'Abstract Figure'. Please also ensure that you include the figure legend in the main article file. All Abstract Figures should be created using BioRender. Authors should use The Journal's premium BioRender account to export high-resolution images. Details on how to use and access the premium account are included as part of this email.

- Please include a full title page as part of your main article (Word) file, which should contain the following: title, authors, affiliations, corresponding author name and contact details, keywords, and running title.

Reviewing Editor's comments:

Your manuscript has been reviewed by three referees, including the Animal Ethics Editor. The reviewers acknowledge the study's potential interest and impact but highlight areas requiring clarification and elaboration. Additional details are needed on ethics and animal welfare, including approval, housing conditions, anaesthesia, surgical procedures, and post-surgical care. Reviewers also raise important issues regarding the interpretation of results, particularly the contraction bias. They suggest alternative explanations for the behavioural and neurophysiological data, such as anchoring effects, and ask for a clearer discussion of evidence supporting one interpretation over another. One reviewer has suggested citing additional

literature on history biases. Please consider whether engaging with this body of work adds to the manuscript's argument and strengthens its contribution to the field.

Referee #1 comments:

Thank you for submitting your manuscript to The Journal of Physiology. Additional details pertaining to animal welfare and ethics are required.

Please consult the most recent update on requirements for ethics and welfare reporting:
<https://physoc.onlinelibrary.wiley.com/doi/10.1113/JP286666>

1. You must start the Methods section with the sub-heading "Ethical approval". Provide the institutional approval code/number for this study.
 2. Provide detail on the source of the animals. If the monkeys were used in previous studies please provide details.
 3. Provide detail on housing arrangements. You must comment on access to food and water.
 4. Line 148/9: How was it determined that an adequate depth of anaesthesia was achieved to permit surgical access?
 5. Provide details on post-surgical care. Was an analgesic given? If not, why not, and confirm that this issue was carefully considered by the approval committee.
 6. Line 153. Give details on the anaesthetic used and how an adequate depth of anaesthesia was determined to allow for thoracotomy. Add that animals were killed/euthanised by this procedure.
 7. Provide details on the training paradigm, and the frequency of training events and general conditions, including details of restraint if used.
 8. The narrative should be organised in a way that better reveals the fate of the animals. Therefore, describe the sequence in the manner that the animal would have experienced so that the fate of the animals is better revealed.
-

Referee #2:

Benozzo et al. examine how recently experienced stimuli can alter behavior and neural representations in a sequential

distance discrimination task. In this task, monkeys were sequentially presented with two objects (S1 and S2) at different distances from the center point, and were rewarded if they chose the image at a greater distance after a brief delay. Interestingly, Monkeys' behavior depended not only on the relative distances of S1 and S2, but on the position of S2 on the previous trial (S2_{previous}): they were more likely to choose S2_{current} when S2_{previous} was small, and less likely to do so when S2_{previous} was large. The authors interpret this as a "contractive bias" affecting S1_{current} - the magnitude of S1 is perceived as more similar to S2_{previous}, shifting the choice pattern after S2 is seen. Interestingly, this effect was stronger when S2_{previous} was a different color than S1_{current}.

To examine the possible neural basis of this result, the authors conducted a decoding analysis on neurons from lateral prefrontal cortex recorded during the task. They found that they could decode the distance S1_{current} starting shortly after image presentation, but that decoding was worse when S2_{previous} had been presented at a different distance than S1_{current}. Decoding was also worse when S1_{current} was the same color as S1_{previous}. The authors also showed that the identity of S2_{previous} could be decoded ~1s after S1 presentation - after the S1 coding effect emerged. Taken together, they interpret these results as evidence that the behavioral effect of S2_{previous} arises from changes in the perception of S1 during stimulus presentation.

Overall the results reported here are interesting and will add to the conversation about history-dependent coding and perception. However, some details of the behavior are hard to interpret, and the decoding results as presented do not strongly support the hypothesis that LPFC neurons mediate a contractive bias.

Major comments:

1) While the decoding analysis suggests that S2_{previous} may affect something about the representation of S1, it does not necessarily show evidence for a contractive/attractive bias. Specifically, the reduced decoding accuracy when different S2_{previous} is different in training and testing suggests a lower signal-to-noise ratio in these trials, perhaps because of greater variability across trials (i.e., some effect of S2_{previous} during S1). However, there is not a strong reason to assume that this effect would lead to a bias in behavior, as opposed to an overall reduction in accuracy or increased choice variability. Notably, decoding performance was also worse when the S2_{previous} is the same color as S1 - an interpretation based on a general "disruption" of the S1 representation would need to account for this as well.

In contrast, if there were a contractive effect on the neural representation, you would expect a similar bias in decoding - e.g., if S1_{current} was "medium" and S2_{previous} was "low", you would expect S1_{current} to be falsely decoded as "low" more often than "high". Is this the case? If not, it is worth interpreting the decoding results with more caution.

2) The effect of S2_{previous} on behavior is interesting, but it is not obvious that it has to be the result of a contractive bias at S1, particularly because there are only two stimuli and they are both used on each trial. For example, since the effect is strongest when S2_{previous} is a different color than S1 (and therefore the same color as S2), it could be related to a type of adaptation or habituation based on object identity - i.e. an effect driven by S2. Alternatively, the identity of S1 could be forgotten or confused with S2_{previous} in some cases, rather than shifted toward it (an effect that might be more obvious when S1 and S2 have greater differences in magnitude). It seems worthwhile to discuss alternate hypotheses in more detail, and include any evidence that would discriminate between one or another.

3) Overall, it is difficult to interpret the behavioral results thoroughly without seeing the monkeys' overall choice pattern. What is the error rate for different pairs? Is there an overall bias towards S1 or S2? Is choice equally sensitive to S1 and S2, or do the monkeys selectively weight the evidence differently (e.g. occasionally ignoring S1 and choosing based on whether S2 is above or below a certain threshold)? Does S2_{previous} shift the psychometric curve, or does it also have an effect on the slope (i.e. more random choices under certain conditions)? Including a psychometric curve or heatmap and a logistic regression to model choice behavior would help clarify the behavioral results and how they may relate to neuronal activity.

Other comments:

1) The authors suggest that habituation may contribute to the effect of color difference on neural and behavioral effects. If so, it seems like there are a few testable predictions. First, is the "reemergence" of S2_{previous} in decoding reduced on same-color trials? Second, is decoding of S1 color reduced on same-color trials, as might be expected if the response to the object is reduced? Third, is the effect of habituation greater when the objects are also at the same location on the screen during presentation (e.g. both on the top half of the screen)?

2) When discussing the decodability of S2_{previous}, the authors describe the previous absence of this signal as a "silent trace." While it is true that they don't see evidence for explicit coding of S2_{previous} in their dataset, this misses possibility that other regions could carry the signal. For example, they discuss several studies of history effects that implicated parietal cortex.

3) Are there differences in eye movements depending on the position of S2previous? E.g. smaller amplitude saccades after a lower-distance S2prev?

4) It's not clear what the HMM analysis adds to the paper. Are there aspects of the state sequence or occupancy that affect the interpretation of results or provide evidence about when and how different information is being processed in this task?

5) Line 37-38: "this occurred only when the stimulus matched the previous visual stimulus features" - this is the opposite of what is stated elsewhere.

6) Lines 100-103: This sentence is hard to parse, and it is unclear whether you mean that the S2prev magnitude is the same for training and testing or the S2prev magnitude is the same as the S1current magnitude.

7) Line 141: "Arrington Recording" should be "Arrington Research"

8) Line 162: Does "six potential magnitudes for each stimulus" refer to the previous trial stimulus? Not totally clear

9) Figure 2 is hard to parse. It takes a while to figure out what the different symbols/colors mean and to determine which parts are data vs. an illustration of the hypothesis. It's also a bit challenging to relate the illustrations of contraction bias (2b and 2a insets) to the data shown in a,c, and d, since they are displayed in a very different way. It might be helpful to rework the illustration and labels to help walk readers through.

10) Parts of the paper state that the contractive effect only occurs when S2previous is different from S1, but based on the results in Fig. 2 it looks like the effect is just noisier/diminished when the objects are the same.

11) Lines 353-354: "there still remains a higher decoding accuracy in the diffColor condition, which aligns with the behavioral data as depicted in Fig. 2(c-d)". This seems like the opposite of what I would expect - if there is more bias, you would expect a less precise representation and therefore worse decoding. What is the logic here? Is it interpreted as an attentional effect?

12) Several tests (e.g. for diffColor vs. sameColor conditions) report a difference that is present in one condition but absent in the other. However, they do not directly test for a difference between conditions, or an interaction term based on color-difference.

13) Line 554: "remind the findings by Histed et al." - I think a word is missing or not right here

14) Lines 602-604: This sentence is hard to parse and a little confusing - wasn't the effect of S2previous on S1 decoding mostly during early rather than late S1?

Dear all,

I have read with care the paper by the Genovesio group and I have the feeling that the paper is on a good track to be published in a venue such as J Physiology, however however some things ought to be improved in order to grant final publication.

1a. One first issues of the work is that it is not tied to all existing literature on History biases. Some findings are mentioned especially those relating to mid-term effects (see Hollingworth, Jazayeri, Cicchini) but others have been skipped. For instance the whole serial dependence saga (with contributions by Whitney, Cicchini, Fritsche, Pascucci) would be very relevant. Interestingly here it is shown that previous stimuli are somewhat laying dormant in the brain and re-emerge with a new stimulation (<https://pubmed.ncbi.nlm.nih.gov/36223998/>). These findings have been also contentious (see here <https://pubmed.ncbi.nlm.nih.gov/31898266/> or here <https://pubmed.ncbi.nlm.nih.gov/33373639/>). So given all this turmoil it would be good if the current data liase to that stream of literature

1b. By a similar note it is worth observing that the human behavioural literature has ascribed an effect similar to the one reported here as an "anchoring effect". The anchoring effects is typically described as the effect of S1 upon S2, however it is not unconceivable that similar mechanisms exists also for S2pre and S1. See here: <https://pubmed.ncbi.nlm.nih.gov/17115044/> or here <https://jov.arvojournals.org/article.aspx?articleid=2541696>. Again it may be worth linking to that literature.

1c. One of the features that is most surprising at behavioural level is that the processing of the previous trial is not over with the previous lab trial but carries forward in the brain. Striking examples of these have been reported by the Mamassian group (see <https://pubmed.ncbi.nlm.nih.gov/22386314/> or <https://pubmed.ncbi.nlm.nih.gov/31251808/>). Perhaps it is also worth citing them.

2. I was surprised by the fact that the behavioural and the decoding analyses do not mirror each other closely. On one hand behavioural data show an attraction of S1 towards S2pre. On the other hand decoding analysis shows that when S2pre is similar to S1 it is easier to decode S1. I have no preconceptions regarding a possible link between a strengthening of a given representation and a possible attractive bias towards it, however what is missing here is the smoking gun of a direct link. Which I think it is a miss given the beauty of such neural recordings. I understand that neural data can be scarce for in depth analyses; I also understand that the main behavioural effect is not creating huge perceptual distortions (a smaller stimulus at S2pre changes by only 5% behaviour responses). However I still believe that something should have been pursued. For instance I would have expected that brain activity after a lower S2pre stimulus would have been a better enhancer of decoding of those trials where S1 was underestimated. I know such an analysis is challenging, but as a reader I would have appreciated that the authors searched a link between the two effects. After all what could be happening here is that the enhancement of decoding for S2pre-S1 pairs at similar eccentricity might be the result of higher firing rates due to the surprise of seeing the similar object presented again (but with a different color). As I said I don't think that such a saliency based explanation is necessarily too plausible, yet I feel the need to disprove such theories.

3. The colors and tags employed in the figures are somewhat hard to follow. For instance why in figure S4a the red and blue labels in the same trials are quite clear to understand. But what about those in the "different color", who is different S2pre from the S1 to which the color refers? It is not too straightforward. Nor the text help too much in that respect.

END OF COMMENTS

Regression model

- 1) Benozzo et al have now added a regression models and model comparisons with reduced versions of the full model:

$$p(\text{left}) = \beta_0 + \beta_1 \cdot \text{stim ratio} + \beta_2 \cdot (S1_{\text{current}} - \langle S \rangle) + \beta_3 \cdot S2_{\text{pre}} + \beta_4 \cdot S1_{\text{pre}}$$

They compare cross-validated R2 across models. It is unclear why they mix regressor formats:

- The stim ratio is a fraction, assuming a non-linear comparison of S1 and S2,
- CB is a relative distance measure (a deviation of the current stimulus from the mean), assuming a linear effect
- S1pre and S2pre are absolute values (not relative to S1_current)

Combining ratio, relative and absolute representations makes the comparison of weights non-straightforward.

For comparison, Akrami et al. use a consistent regressor format, which allows the corresponding coefficients to be compared directly.

The authors should present a regression model with consistent coding of variables.

- 2) Relatedly, the current comparison of weights suggests that the decision to choose left is in fact more strongly driven by the CB on S1 than by the actual stimulus ratio (as also stated in Benozzo 2023). I wonder whether this result remains valid when a consistent regressor format (e.g. Akrami et al) is chosen.
- 3) Similarly, the results suggest that the CB explains more variance than bias towards S2, but a consistent choice of coding should be presented to support this claim.
- 4) Taken together, the authors should state more clearly in the results section, just before going to the neural results section, that the main effect in their behavioral results stems from CB, but that they nonetheless focus on the neural correlates of the weaker S2_prev-effect for practical reasons and reasoning in the previous literature.

Interpretation of reactivation of “activity-silent” codes

- 1) I hadn't understood that the authors interpreted the interaction effect of S2_prev x S1_current on S1 decoding as “reactivation from a silent trace”. Instead, I had read Fig 4a,c as S1 encoding being biased towards S2_prev. It is not clear from the analysis whether the decoding is the result of one versus the other.

A more straightforward analysis is the decoding of S2_prev throughout the current trial, as in Barbosa et al., and as reported in Fig 6d. I suggest moving 6d alongside Fig 4a,c so that the interpretation in terms of S2 reactivation is more clearly interpreted. Please at minimum state clearly in the results that you interpret the S1 decoding bias as a reactivation, so that the discussion is easier to follow.

Decoding states/HMM

- 2) “After estimating the state sequence, the presence of a coding state - defined as a state that codes for a specific task variable - was evaluated by comparing the mean state occupancy between subsets of trials grouped according to the task variable under consideration”

This is not a precise definition. I understand that this analysis has been used before, but

methods for the current paper should still be sufficiently tractable without having to search through the authors' publication history.

- 3) I still think the HMM analysis is overly complex for what the authors want to show: that there are neurons whose firing patterns during S1 presentation is described by firing rate = condition x color , and that these neurons don't show the same interaction effect during pre-stimulus. If the authors want to keep the HMM analysis, I suggest they streamline the HMM results description, get rid of overly complicated introductions to the technique/previous results etc. and highlight what we learn.

Taken together, I think the manuscript can be published if the above comments are addressed for interpretability and readability.

JP-RP-2024-288070 - The Journal of Physiology - Decision Letter

REQUIRED ITEMS

- Author photo and profile. First or joint first authors are asked to provide a short biography (no more than 100 words for one author or 150 words in total for joint first authors) and a portrait photograph. These should be uploaded and clearly labelled together in a Word document with the revised version of the manuscript. See Information for Authors for further details.

Done

- You must start the Methods section with a paragraph headed Ethical approval (https://jp.msubmit.net/cgi-bin/main.plex?form_type=display_requirements#methods).

Done, see from line 137

Research must comply with The Journal's policies regarding animal experiments (<https://physoc.onlinelibrary.wiley.com/hub/animal-experiments>) and adherence to these policies must be stated in the manuscript.

Authors should confirm in their Methods section that their experiments were carried out according to the guidelines laid down by their institution's animal welfare committee, including an ethics approval reference number. The Methods section must contain a statement about access to food, water and housing, details of the anaesthetic regime: anaesthetic used, dose and route of administration, and method of killing the experimental animals.

Done, see subsection "Animals" line 146

- Your manuscript must include a complete Additional Information section, including competing interests; funding; author contributions and acknowledgements.

Done, see from line 791

- Please upload separate high-quality figure files via the submission form.

Ok, done

- Your paper contains Supporting Information of a type that we no longer publish, including supplementary tables and figures. Any information essential to an understanding of the paper must be included as part of the main manuscript and figures. The only Supporting Information that we publish are video and audio, 3D structures, program codes and large data files. Your revised paper will be returned to you if it does not adhere to our Supporting Information Guidelines.

Now the "supporting informations" section has been removed, all figures in that section have been included in the manuscript

- Papers must comply with the Statistics Policy: https://jp.msubmit.net/cgi-bin/main.plex?form_type=display_requirements#statistics.

In summary:

- If n {less than or equal to} 30, all data points must be plotted in the figure in a way that reveals their range and distribution. A bar graph with data points overlaid, a box and whisker plot or a violin plot (preferably with data points included) are acceptable formats.
- If $n > 30$, then the entire raw dataset must be made available either as supporting information, or hosted on a not-for-profit repository, e.g. FigShare, with access details provided in the manuscript.
- 'n' clearly defined (e.g. x cells from y slices in z animals) in the Methods. Authors should be mindful of pseudoreplication.
- All relevant 'n' values must be clearly stated in the main text, figures and tables.
- The most appropriate summary statistic (e.g. mean or median and standard deviation) must be used. Standard Error of the Mean (SEM) alone is not permitted.
- Exact p values must be stated. Authors must not use 'greater than' or 'less than'. Exact p values must be stated to three significant figures even when 'no statistical significance' is claimed.

Done, exact p-values are reported for key results. Data are presented using boxplots, or by plotting all individual data points. In cases where histograms are used, the standard deviation is also shown; however, histograms are always supplementary visualization of data previously presented as boxplots, e.g. Fig.4a. For decoding curves, we reported the mean and SD along with the corresponding distribution statistics.

- Please include an Abstract Figure file, as well as the Figure Legend text within the main article file. The Abstract Figure is a piece of artwork designed to give readers an immediate understanding of the research and should summarise the main conclusions. If possible, the image should be easily 'readable' from left to right or top to bottom. It should show the physiological relevance of the manuscript so readers can assess the importance and content of its findings. Abstract Figures should not merely recapitulate other figures in the manuscript. Please try to keep the diagram as simple as possible and without superfluous information that may distract from the main conclusion(s). Abstract Figures must be provided by authors no later than the revised manuscript stage and should be uploaded as a separate file during online submission labelled as File Type 'Abstract Figure'. Please also ensure that you include the figure legend in the main article file. All Abstract Figures should be created using BioRender. Authors should use The Journal's premium BioRender account to export high-resolution images. Details on how to use and access the premium account are included as part of this email.

Done, Abstract Figure file is both in the main article file and separately uploaded, the Figure Legend text starts from line 38.

- Please include a full title page as part of your main article (Word) file, which should contain the following: title, authors, affiliations, corresponding author name and contact details, keywords, and running title.

Done

Editor's comments

Your manuscript has been reviewed by three referees, including the Animal Ethics Editor. The reviewers acknowledge the study's potential interest and impact but highlight areas requiring clarification and elaboration. Additional details are needed on ethics and animal welfare, including approval, housing conditions, anaesthesia, surgical procedures, and post-surgical care. Reviewers also raise important issues regarding the interpretation of results, particularly the contraction bias. They suggest alternative explanations for the behavioural and neurophysiological data, such as anchoring effects, and ask for a clearer discussion of evidence supporting one interpretation over another. One reviewer has suggested citing additional literature on history biases. Please consider whether engaging with this body of work adds to the manuscript's argument and strengthens its contribution to the field.

Dear Editor,

We want to thank the reviewers for carefully reading and evaluating our manuscript.

Please find below our point-to-point reply to the reviewers' questions (line numbers refer to the "tracked changes" version of the manuscript),

REFEREE COMMENTS

Referee #1 (ethics review)

Thank you for submitting your manuscript to The Journal of Physiology. Additional details pertaining to animal welfare and ethics are required.

Please consult the most recent update on requirements for ethics and welfare reporting:
<https://physoc.onlinelibrary.wiley.com/doi/10.1113/JP286666>

1. You must start the Methods section with the sub-heading "Ethical approval". Provide the institutional approval code/number for this study.

We added the sub-heading "Ethical Approval" at the beginning of the Methods section, lines (130-136):

"The current study is a data analysis study based on experimental recordings reported in previously published works. At the time recordings were obtained, all animal surgical and experimental procedures were approved in advance by the National Institute of Mental Health Animal Care and Use Committee [LSN_03_05] and followed the National Institutes of Health Guide for the Care and Use of Laboratory Animals (1996) and were approved by the National Institute of Mental Health Animal Care and Use Committee. The authors understand the ethical principles under which The Journal of Physiology operates and confirm that this work complies with its animal ethics checklist."

2. Provide detail on the source of the animals. If the monkeys were used in previous studies please provide details.

Unfortunately, we do not have access to information about the origin of the animals, since previous articles that used the same task and the same animals do not report this information. We reported the articles that were published in the past using the same task/monkeys in the *Animals* subsection, *Methods* section (lines 147-150): ***"The information about the care and welfare of the animals is as reported in previously published articles that used the same task/animals (Genovesio et al. 2009, 2011, 2012, 2015, 2016; Marcos et al. 2017, 2019; Benozzo et al. 2021, Benozzo et al. 2023)."***

3. Provide detail on housing arrangements. You must comment on access to food and water.

We reported the information about husbandry and monkey's diet in the *Animals* subsection, *Methods* section (lines 141-147): ***"To encourage participation during training and neural recordings, the animals' water intake was regulated. Monkeys had unrestricted access to dry food. Following each day's experimental sessions, they were given fresh foods like fruits and vegetables. Body weight was measured several times per week and maintained at no less than 85% of the pre-water-control baseline weight. Full-time, on-site veterinary personnel closely monitored the animals' weight and overall health. Monkeys were housed in pairs unless temporary separation was necessary due to adverse outcomes."***

4. Line 148/9: How was it determined that an adequate depth of anaesthesia was achieved to permit surgical access?

All the surgical procedures were conducted under the supervision of responsible veterinary staff. We added this sentence in the *Surgery and histological analysis* subsection, *Methods* section (lines 196-197): **“The entire euthanasia procedure was conducted under extremely deep anaesthesia and under the supervision of the responsible on-site veterinary staff.”**

5. Provide details on post-surgical care. Was an analgesic given? If not, why not, and confirm that this issue was carefully considered by the approval committee. Unfortunately, we do not have access to veterinary treatment protocols, since previous articles that used the same task and the same animals did not report this information. We added this statement in the *Surgery and histological analysis* subsection, *Methods* section (lines 190-192): **“After surgery, the monkeys were monitored daily by full-time on site veterinary staff who managed the use of therapies and analgesics to ensure full post-operative recovery and the general welfare of the animals.”**

6. Line 153. Give details on the anaesthetic used and how an adequate depth of anaesthesia was determined to allow for thoracotomy. Add that animals were killed/euthanised by this procedure. Unfortunately, we do not have access to veterinary procedures during euthanasia, since previous articles that used the same task and the same animals do not report this information. We added this statement in the *Surgery and Histological Analysis* subsection, *Methods* section (lines XX-XX): **“The entire euthanasia procedure was conducted under extremely deep anaesthesia and under the supervision of the on-site veterinary staff”**

7. Provide details on the training paradigm, and the frequency of training events and general conditions, including details of restraint if used. We reported the information about training regime in the *Animals* subsection, *Methods* section (lines 139-141): **“The monkeys were trained prior to surgery and the start of recordings for a period of approximately 2 years, on a monday to friday schedule with two resting days during the weekend.”**

8. The narrative should be organised in a way that better reveals the fate of the animals. Therefore, describe the sequence in the manner that the animal would have experienced so that the fate of the animals is better revealed. We added this sentence to clarify the fate of the animals in the *Surgery and Histological Analysis* subsection, *Methods* section (lines 196-197): **“The entire euthanasia procedure was conducted under extremely deep anaesthesia and under the supervision of the on-site veterinary staff”**

Referee #2

Benozzo et al. examine how recently experienced stimuli can alter behavior and neural representations in a sequential distance discrimination task. In this task, monkeys were sequentially presented with two objects (S1 and S2) at different distances from the center point, and were rewarded if they chose the image at a greater distance after a brief delay. Interestingly, Monkeys' behavior depended not only on the relative distances of S1 and S2,

but on the position of S2 on the previous trial (S2previous): they were more likely to choose S2current when S2previous was small, and less likely to do so when S2previous was large. The authors interpret this as a "contractive bias" affecting S1current - the magnitude of S1 is perceived as more similar to S2previous, shifting the choice pattern after S2 is seen. Interestingly, this effect was stronger when S2previous was a different color than S1current.

To examine the possible neural basis of this result, the authors conducted a decoding analysis on neurons from lateral prefrontal cortex recorded during the task. They found that they could decode the distance S1current starting shortly after image presentation, but that decoding was worse when S2previous had been presented at a different distance than S1current. Decoding was also worse when S1current was the same color as S1previous. The authors also showed that the identity of S2previous could be decoded ~1s after S1 presentation - after the S1 coding effect emerged. Taken together, they interpret these results as evidence that the behavioral effect of S2previous arises from changes in the perception of S1 during stimulus presentation.

Overall the results reported here are interesting and will add to the conversation about history-dependent coding and perception. However, some details of the behavior are hard to interpret, and the decoding results as presented do not strongly support the hypothesis that LPFC neurons mediate a contractive bias.

Major comments:

1) While the decoding analysis suggests that S2previous may affect something about the representation of S1, it does not necessarily show evidence for a contractive/attractive bias. Specifically, the reduced decoding accuracy when different S2previous is different in training and testing suggests a lower signal-to-noise ratio in these trials, perhaps because of greater variability across trials (i.e., some effect of S2previous during S1). However, there is not a strong reason to assume that this effect would lead to a bias in behavior, as opposed to an overall reduction in accuracy or increased choice variability. Notably, decoding performance was also worse when the S2previous is the same color as S1 - an interpretation based on a general "disruption" of the S1 representation would need to account for this as well.

Thank you for highlighting this point.

A contraction/attraction of the actual S1 magnitude encoding towards S2previous can help explain the behavioral effect shown in Fig. 2a within the framework of contraction bias. This interpretation builds on findings from Akrami et al., 2018, Raviv et al., 2012, Preuschhof et al., 2010 etc. We agree that our neuronal analyses alone are not sufficient to draw definitive conclusions. Our decoding results are intended to provide supporting evidence for an altered encoding of S1 magnitude influenced by recent history, i.e. S2previous, rather than to directly demonstrate the presence of an attraction effect (for new results on this see also the next response). Furthermore, the observed distortion in S1 encoding due to recent history supports the view that the history effect is not an independent bias contributing to behaviour, but rather operated within the framework of contraction bias.

In contrast, if there were a contractive effect on the neural representation, you would expect a similar bias in decoding - e.g., if S1current was "medium" and S2previous was "low", you would expect S1current to be falsely decoded as "low" more often than "high". Is this the case? If not, it is worth interpreting the decoding results with more caution.

Thank you for this suggestion. Following this line of reasoning, we examined how the marginal probability of each class varied when decoders trained on the two extreme cases, S2previous low and high, were applied to decode S1 on trials with a different history. More specifically, given a decoder trained to discriminate S1 in trials with low S2previous, we computed the marginal probabilities of assigning a low, medium, or high label to trials with the same S2previous. These probabilities serve as a reference, as they are not influenced by history bias (training and testing trials share the same S2previous category). We then applied the same decoder to trials with medium and high S2previous, and computed how the marginal probabilities deviated from the reference case. In this way, we quantified whether the decoder's predictions were biased when applied to trials with an S2previous that differed from the one it was trained on. Indeed, it emerged that the decoder's predictions were biased towards its own training S2previous. Similarly when trained on high S2previous and tested on low and medium S2previous.

It is important to emphasize that this analysis is not intended to reproduce the neural mechanisms underlying the history bias in S1 encoding. Rather, it shows that a linear decoder exhibits a similar bias that could account for the observed behavioral results. Fig.4 has been updated to include the variations of S1 marginal probabilities for each trial class (S1 low, medium, and high), when the decoder was trained on low and high S2previous. These variations are shown across two time windows: earlyS1 (panel b), and lateS1 (panel d).

2) The effect of S2previous on behavior is interesting, but it is not obvious that it has to be the result of a contractive bias at S1, particularly because there are only two stimuli and they are both used on each trial. For example, since the effect is strongest when S2previous is a different color than S1 (and therefore the same color as S2), it could be related to a type of adaptation or habituation based on object identity - i.e. an effect driven by S2. Alternatively, the identity of S1 could be forgotten or confused with S2previous in some cases, rather than shifted toward it (an effect that might be more obvious when S1 and S2 have greater differences in magnitude). It seems worthwhile to discuss alternate hypotheses in more detail, and include any evidence that would discriminate between one or another.

When considering the S2-ward bias effect in Fig.2a, its interpretation within the framework of contraction bias aligns with previous studies on magnitude discrimination tasks. Specifically, it suggests that recent history influences the estimate of the mean stimulus magnitude, thus resulting in an altered encoding of the S1 value. The method used to compute the S2-ward bias also follows prior work (Akrami et al., 2018), where the bias is defined as the deviation from the average performance for a given stimulus pair (S1,S2), regardless of its history. This approach helps isolate the effect of S2previous by factoring out potential biases unrelated to it. Moreover, since the defining features of a stimulus (shape, color, and position) are always complementary within a trial, Fig.S2 which considers the effect of S1previous, was intended to rule out the potential contribution of other biases/tendencies, such as a color shift. To corroborate this, we computed the Sred-ward bias, analogous to the S2-ward bias but based on stimulus color rather than presentation order. Specifically, we quantified the bias toward choosing the red stimulus (Sred), using the value of the previous Sred (left panel) and the previous Sblue (right panel) to group the last seen trial.

Figure R1. Effect of previous stimulus distance on the performance. In the left panel, the magnitude of the red stimulus (Sred) from the previous trial was considered; in the right panel, the magnitude of the blue stimulus (Sblue) was used. In both cases, the bias is quantified as the tendency to respond to Sred (Sred-ward bias).

From Fig.R1, we see that in both cases the bias ranges around -1 to +1%, which is smaller than the S2-ward bias. More importantly, the two biases cancel each other out when considering both colors of the previous stimulus pair, suggesting a form of habituation or preference in the animal's choice. This pattern does not hold for the S2-ward bias.

3) Overall, it is difficult to interpret the behavioral results thoroughly without seeing the monkeys' overall choice pattern. What is the error rate for different pairs? Is there an overall bias towards S1 or S2? Is choice equally sensitive to S1 and S2, or do the monkeys selectively weight the evidence differently (e.g. occasionally ignoring S1 and choosing based on whether S2 is above or below a certain threshold)? Does S2previous shift the psychometric curve, or does it also have an effect on the slope (i.e. more random choices under certain conditions)? Including a psychometric curve or heatmap and a logistic regression to model choice behavior would help clarify the behavioral results and how they may relate to neuronal activity.

The error rate for each pair is reported in Fig.R2, middle panel. It reflects the contraction bias expectation (left panel) and indeed it has been the core of our previous work on this topic (Benozzo et al., 2023). As mentioned on the response 2, the specific error rate of each pair (S1, S2) was factored out when computing the S2-ward bias thus it does not affect the interpretation of the effect of S2previous. Fig.R2 also shows the psychometric curves representing the probability that a trial had $S2 > S1$, given that an error occurred, focusing on stimulus pairs with the smallest difference, i.e. 8 mm (right panel). The results indicate that the stimulus pairs most strongly affected by the S2-ward bias are those where the contraction bias has a negative impact. For example, when S2previous was low (producing a positive S2-ward bias) trials with low mean magnitude and $S2 > S1$ (e.g. pair (8,16), bottom left corner of the first panel) benefit from the bias, while trials where $S1 > S2$ and high mean magnitude (top right corner) are negatively impacted. As a result, the corresponding curve (right panel) shifts downward relative to the average. The opposite occurs when S2previous was high, which biases choices toward S1, leading to a negative effect on pairs like (8,16) and a positive effect on pairs like (48,40), resulting in an upward shift of the psychometric curve.

Figure R2. Expected bias due to contraction bias (left panel). Performance accuracy of each stimulus pair (middle panel). Effect of the previous S2 on the performance of the current trial for stimulus pairs (S1,S2) with the smallest magnitude difference, $|S2-S1|=8\text{mm}$, grouped by low, medium and high S2previous magnitude (left panel).

An alternative representation, to visualise how S2previous alters the performance is also reported here in Fig.R3 (Fig.S1 in the original manuscript, now simplified in Fig. 2(e,f)). In particular, it shows the S2-ward bias for each stimulus current pair (S1,S2) by dividing trials according to the previous S2 (low S2previous on the left panel, and high S2previous on the right panel).

Figure R3. Effect of previous S2 on the performance of the current trial for each stimulus current pair (S1,S2). Trials have been divided according to the value of the previous S2, in the panel only trials with low values of previousS2 (8 or 16 mm), in the panel b only trials with high S2 previous (40 or 48 mm).

Other comments:

1) The authors suggest that habituation may contribute to the effect of color difference on neural and behavioral effects. If so, it seems like there are a few testable predictions. First, is the "reemergence" of S2previous in decoding reduced on same-color trials? Second, is decoding of S1 color reduced on same-color trials, as might be expected if the response to the object is reduced? Third, is the effect of habituation greater when the objects are also at the same location on the screen during presentation (e.g. both on the top half of the screen)?

- Regarding the decoding of S2previous in same/diffColor trials, Fig.R4 reports the two corresponding decoding curves. A reduction in decoding accuracy is observed in the sameColor condition (left panel). In the diffColor case (right panel), a peak in decoding accuracy is still present, but it occurs earlier than the peak shown in Fig.6d of the manuscript where all trials were used. The reason for this temporal shift is not easy to interpret, especially considering that splitting the data into same/diffColor conditions halves the number of trials available for the decoding analysis, thus reducing its statistical power.

Figure R4. Decoding of previous S2 magnitude (low, medium, and high) during the presentation window of S1 (current trial), specifically from 400 ms before S1 appearance (0 ms) until 400 ms after the end of its presentation period (which lasted for 1000 ms, the final 400 ms is the shortest delay D1 in common with all trials). Left panel refers to the diffColor condition, while the right panel to the sameColor one.

- decoding of S1 color was already reported in Fig.6(a-c), in particular panels b and c differentiate between same and diff color, reporting a reduction on same color trials;
- regarding stimulus location on the screen, Fig.R5 shows the S2-ward bias separately for trials in which the positions of S2previous and S1current differed (left panel), and for trials where they were the same (right panel). In the different-position condition, the slope of the linear fit is -1.64 ($p=0.002$), while in the same-position condition, the slope is -1.02 ($p=0.0006$), (added in the manuscript, see lines 389-394). Since both slopes are significantly not null, we computed a two-way ANOVA with two independent variables: the feature (color vs. position), and the coherence between S2previous and S1 current (same vs. different). It results that coherence was the only significant factor ($p=0.0002$). The results of the post-hoc test are as follows

Multiple Comparison of Means - Tukey HSD,FWER=0.05

```

=====
group1 group2 meandiff lower upper reject
-----
0.0 1.0 -1.0564 -2.6432 0.5303 False
0.0 2.0 0.5833 -1.0035 2.1701 False
0.0 3.0 -1.6391 -3.2259 -0.0524 True
1.0 2.0 1.6397 0.053 3.2265 True
1.0 3.0 -0.5827 -2.1695 1.004 False
2.0 3.0 -2.2225 -3.8092 -0.6357 True
=====

```

where 0: diffPosition, 1: samePosition, 2: diffColor, and 3: sameColor. In summary, the results show that the sameColor condition yielded an S2-ward bias that was significantly different from both the diffColor ($p=0.002$) and diffPosition (0.04) conditions. Additionally, the samePosition condition differed significantly from the diffColor one ($p=0.04$). However, for the position feature, the difference between same and different conditions was not statistically significant. It is important to note that, unlike color, position is influenced by the distance of the stimulus, e.g. a stimulus 8 mm above the center is closer to one located 8 mm below the center than to one 48 mm above the center. This can explain why color, which is a strictly categorical feature, has a stronger effect on modulating the S2-ward bias.

Figure R5. Effect of the previous S2 distance on the performance. In the left panel only trials where S2previous and S1current had different screen positions are included; in the right panel, only trials with the same position are shown. In both cases, the S2-ward bias is plotted as a function of the S2previous distance.

2) When discussing the decodability of S2previous, the authors describe the previous absence of this signal as a "silent trace." While it is true that they don't see evidence for explicit coding of S2previous in their dataset, this misses possibility that other regions could carry the signal. For example, they discuss several studies of history effects that implicated parietal cortex.

Yes, we agree. Reporting a "silent trace" effect based on the analysis of recordings from a specific set of cells does not allow for generalizing the effect beyond the population under study, and therefore does not exclude the possibility that other regions may also be involved in the encoding (added in the Discussion, lines 710-713).

3) Are there differences in eye movements depending on the position of S2previous? E.g. smaller amplitude saccades after a lower-distance S2prev?

Although eye movements were recorded, this task did not require eye fixation. As a result, saccades during stimulus presentation are not informative, given the absence of a fixed reference point. Moreover, the animals were not trained to use their eye movements as their interaction with the task was mediated through touching switches, which represented the monkey motor response, rather than through gaze-dependent responses.

4) It's not clear what the HMM analysis adds to the paper. Are there aspects of the state sequence or occupancy that affect the interpretation of results or provide evidence about when and how different information is being processed in this task?

The HMM analysis was intended to support the decoding results of the color condition (same/diffColor) by studying the presence of coding states associated with this variable. Unlike the decoding shown in Fig.6(a-c) which is a population-level analysis performed on a pseudo-dataset constructed by combining data across sessions, the HMM analysis is more localized, operating at the level of individual sessions (lines 545-546). This was feasible because the color condition is a binary variable, which avoids the need to divide trials into multiple classes. As a result, it provided a good compromise between the number of available trials per session and the optimal working condition of the HMM analysis.

5) Line 37-38: "this occurred only when the stimulus matched the previous visual stimulus features" - this is the opposite of what is stated elsewhere.

Corrected

6) Lines 100-103: This sentence is hard to parse, and it is unclear whether you mean that the S2prev magnitude is the same for training and testing or the S2prev magnitude is the same as the S1current magnitude.

Done

7) Line 141: "Arrington Recording" should be "Arrington Research"

Done

8) Line 162: Does "six potential magnitudes for each stimulus" refer to the previous trial stimulus? Not totally clear

Yes, this refers to the S2 magnitude of the previous trial. Done, line 205-206.

9) Figure 2 is hard to parse. It takes a while to figure out what the different symbols/colors mean and to determine which parts are data vs. an illustration of the hypothesis. It's also a bit challenging to relate the illustrations of contraction bias (2b and 2a insets) to the data shown in a,c, and d, since they are displayed in a very different way. It might be helpful to rework the illustration and labels to help walk readers through.

Figure 2 has been revised and also a more detailed description has been added.

10) Parts of the paper state that the contractive effect only occurs when S2previous is different from S1, but based on the results in Fig. 2 it looks like the effect is just noisier/diminished when the objects are the same.

We have now clarified this point in the manuscript. In the behavioral data, we observed a reduction in the slope, which is significantly different from that of the different-color condition (test added in line 355-358). From the neural analysis, the same-color condition showed both a reduced ability to decode the S1 magnitude and a weaker dependence on the S2 previous value (how these two phenomena are related is beyond the scope of this study, we are not in a position to make claims about their link or possible causal interaction between them).

11) Lines 353-354: "there still remains a higher decoding accuracy in the diffColor condition, which aligns with the behavioral data as depicted in Fig. 2(c-d)". This seems like the opposite of what I would expect - if there is more bias, you would expect a less precise representation and therefore worse decoding. What is the logic here? Is it interpreted as an attentional effect?

The purpose of Fig.3 was to focus exclusively on the decoding of S1, independently of the previous trial. The logic was to first confirm that S1 could be reliably decoded, as a prerequisite for later evaluating the effect of S2previous on S1 decoding. Since the results shown in Fig.3(c,d), are based on decoders trained within each color condition, it is expected that regardless of any bias the decoder will optimize its decision rule to perform best within that specific condition.

However, it is difficult to determine whether the observed reduction in S1 decoding accuracy in the sameColor condition is due to history bias, the color match itself, or an unrelated factor. Nevertheless, we have some indications that history bias is less likely to be the main contributor at this stage. In particular, S1 decoding performance was preserved in the diffColor condition, where behavioural data suggest a stronger history effect. Conversely, the color condition seems more likely to play a role, since S1 color decoding was also impaired in the sameColor case, Fig.6(b,c). Alternatively, there may be a third factor specifically

affecting the sameColor condition, which impairs both S1 magnitude decoding and S1 color decoding, and also contributes to the S2-ward bias. Beyond these considerations, it is also important to acknowledge that splitting trials by color condition reduces the number of trials per group, which inevitably impacts the statistical power of the decoding analysis.

12) Several tests (e.g. for diffColor vs. sameColor conditions) report a difference that is present in one condition but absent in the other. However, they do not directly test for an difference between conditions, or an interaction term based on color-difference.

These tests have now been added to the behavioural results regarding the S2-ward bias, and to the new results reported in the updated version of Fig.4 panel (e).

13) Line 554: "remind the findings by Histed et al." - I think a word is missing or not right here
Done, line 686

14) Lines 602-604: This sentence is hard to parse and a little confusing - wasn't the effect of S2previous on S1 decoding mostly during early rather than late S1?

We rephrased the sentence, lines 742-745.

Referee #3

Dear all,

I have read with care the paper by the Genovesio group and I have the feeling that the paper is on a good track to be published in a venue such as J Physiology, however however some things ought to be improved in order to grant final publication.

1a. One first issues of the work is that it is not tied to all existing literature on History biases. Some findings are mentioned especially those relating to mid-term effects (see Hollingworth, Jazayeri, Cicchini) but others have been skipped. For instance the whole serial dependence saga (with contributions by Whitney, Cicchini, Fritsche, Pascucci) would be very relevant. Interestingly here it is shown that previous stimuli are somewhat laying dormant in the brain and re-emerge with a new stimulation (<https://pubmed.ncbi.nlm.nih.gov/36223998/>). These findings have been also contentious (see here <https://pubmed.ncbi.nlm.nih.gov/31898266/> or here <https://pubmed.ncbi.nlm.nih.gov/33373639/>). So given all this turmoil it would be good if the current data liase to that stream of literature

Thank you for suggesting these works, they have been mentioned in lines 733-737:

"In this line, Ranieri et al. (2022) reported the activation of an activity-silent trace related to the previous response by applying decoding analysis on EEG recording during a spatial frequency discrimination task. Notably, the peak in decoding accuracy occurred 700 ms after the presentation of the current stimulus, similar to the time window in which we observed the reactivation of the previous S2."

and lines 755-761:

"The fact that visual stimulus features influenced the serial bias on the following trial, whereas the spatial feature (i.e., whether the stimulus was shown above or below the central target) did not, echoes the findings of Bae et al. (2020). In their study, serial dependence selectively affected the feature that had been reported in the previous trial, which in our task always corresponds to the visual one."

1b. By a similar note it is worth observing that the human behavioural literature has ascribed an effect similar to the one reported here as an "anchoring effect". The anchoring effects is typically described as the effect of S1 upon S2, however it is not unconceivable that similar mechanisms exists also for S2pre and S1. See here:

<https://pubmed.ncbi.nlm.nih.gov/17115044/> or here

<https://jov.arvojournals.org/article.aspx?articleid=2541696>. Again it may be worth linking to that literature.

Done, lines 72-76

"Interestingly, the extent to which this bias alters stimulus perception has been found to be smaller in individuals with dyslexia, both in terms of contraction toward the stimulus mean (during auditory discrimination) and serial dependence (during visual discrimination) (Ahissar et al., 2006; Jaffe-Dax et al., 2016). This has been interpreted as a reduced ability to integrate stimulus statistics, consequently affecting the acquisition of reading skills."

1c. One of the features that is most surprising at behavioural level is that the processing of the previous trial is not over with the previous lab trial but carries forward in the brain.

Striking examples of these have been reported by the Mamassian group (see

<https://pubmed.ncbi.nlm.nih.gov/22386314/> or <https://pubmed.ncbi.nlm.nih.gov/31251808/>).

Perhaps it is also worth citing them.

Done, line 64

"Previous experience influences and biases our behavior (Maloney et al., 2005; Gekas et al., 2019)."

2. I was surprised by the fact that the behavioural and the decoding analyses do not mirror each other closely. On one hand behavioural data show an attraction of S1 towards S2pre. On the other hand decoding analysis shows that when S2pre is similar to S1 it is easier to decode S1. I have no preconceptions regarding a possible link between a strengthening of a given representation and a possible attractive bias towards it, however what is missing here is the smoking gun of a direct link. Which I think it is a miss given the beauty of such neural recordings. I understand that neural data can be scarce for in depth analyses; I also understand that the main behavioural effect is not creating huge perceptual distortions (a smaller stimulus at S2pre changes by only 5% behaviour responses). However I still believe that something should have been pursued. For instance I would have expected that brain activity after a lower S2pre stimulus would have been a better enhancer of decoding of those trials where S1 was underestimated. I know such an analysis is challenging, but as a reader I would have appreciated that the authors searched a link between the two effects. After all what could be happening here is that the enhancement of decoding for S2pre-S1 pairs at similar eccentricity might be the result of higher firing rates due to the surprise of seeing the similar object presented again (but with a different color). As I said I don't think that such a saliency based explanation is necessarily too plausible, yet I feel the need to disprove such theories.

Thank you for raising this point, which indeed aligns with the first question from Reviewer 2. In short, to better understand why the condition of similar values between Spre and S1 facilitates S1 decoding, we focused on the marginal probabilities of predicting each of the three S1 classes (low, medium, and high). This approach provides a way to quantify potential bias in the decoder's output, which may not be directly reflected in the overall decoding accuracy. For more details, please refer to the updated Fig. 4 (in particular panels b and d), lines 489-504 of the manuscript, and our response to Reviewer 2's first question.

3. The colors and tags employed in the figures are somewhat hard to follow. For instance why in figure S4a the red and blue labels in the same trials are quite clear to understand. But what about those in the "different color", who is different S2pre from the S1 to which the color refers? It is not too straightforward. Nor the text help too much in that respect.

Fig. S4 (now included in Fig. 5) shows in each panel the temporal activation of a single state which was identified as a coding state for the color variable, but only within one specific condition (either sameColor or diffColor). For example, in the top-left panel, state_0 clearly distinguishes between red and blue labels only in the sameColor trials. This indicates that the state encodes color information specifically under the sameColor condition, and not in the diffColor trials.

Dear Dr Genovesio,

Re: JP-RP-2025-288070R1 "History bias and its perturbation of the stimulus representation in the macaque prefrontal cortex." by Danilo Benozzo, Lorenzo Ferrucci, Francesco Ceccarelli, and Aldo Genovesio

Thank you for submitting your manuscript to The Journal of Physiology. It has been assessed by a Reviewing Editor and by 2 expert referees and we are pleased to tell you that it is potentially acceptable for publication following satisfactory major revision.

REVISION CHECKLIST:

Please upload two versions of your manuscript text: one with all relevant changes highlighted and one clean version with no

changes tracked. The manuscript file should include all tables and figure legends, but each figure/graph should be uploaded as separate, high-resolution files.

We look forward to receiving your revised submission.

Yours sincerely,

Richard Carson
Senior Editor
The Journal of Physiology

REQUIRED ITEMS

1) - You must start the Methods section with a paragraph headed Ethical approval (https://jp.msubmit.net/cgi-bin/main.plex?form_type=display_requirements#methods).

Research must comply with The Journal's policies regarding animal experiments (<https://physoc.onlinelibrary.wiley.com/hub/animal-experiments>) and adherence to these policies must be stated in the manuscript.

Authors should confirm in their Methods section that their experiments were carried out according to the guidelines laid down by their institution's animal welfare committee, including an ethics approval reference number. The Methods section must contain a statement about access to food, water and housing, details of the anaesthetic regime: anaesthetic used, dose and route of administration, and method of killing the experimental animals.

Specifically, please include:

- Details of housing/ environment (light/dark cycle, temperature)
- The agent, route and dose of anaesthesia
- More details on termination/ euthanasia.

2) - Papers must comply with the Statistics Policy: https://jp.msubmit.net/cgi-bin/main.plex?form_type=display_requirements#statistics.

In summary:

- If $n \leq 30$, all data points must be plotted in the figure in a way that reveals their range and distribution. A bar graph with data points overlaid, a box and whisker plot or a violin plot (preferably with data points included) are acceptable formats.
- If $n > 30$, then the entire raw dataset must be made available either as supporting information, or hosted on a not-for-profit repository, e.g. FigShare, with access details provided in the manuscript.
- 'n' clearly defined (e.g. x cells from y slices in z animals) in the Methods. Authors should be mindful of pseudoreplication.
- All relevant 'n' values must be clearly stated in the main text, figures and tables.
- The most appropriate summary statistic (e.g. mean or median and standard deviation) must be used. Standard Error of the Mean (SEM) alone is not permitted.
- Exact p values must be stated. Authors must not use 'greater than' or 'less than'. Exact p values must be stated to three significant figures even when 'no statistical significance' is claimed. $p < 0.05$ is not acceptable, exact p values must be used.

EDITOR COMMENTS

Thank you for your thorough and thoughtful revision. Both reviewers agreed that the revised manuscript is much improved and that the data are of interest. Key points raised include the need to clarify the interpretation of the behavioural effect (e.g., contraction bias vs. 1-back influence), provide a more balanced discussion of alternative hypotheses, and reconsider the relevance of the HMM analysis.

Senior Editor:

Please refer to the Statistical Policy of the Journal in respect of the use of asterisks on figures to indicate "statistically significant" values. For example, in the legend to figure 4, it is written "ANOVA test with Tukey's multiple comparison test".

In this regard also, please provide the test statistic (e.g., F ratio, Mann-Whitney U) and associated degrees of freedom in all cases (i.e., not just the exact p value).

REFEREE COMMENTS

Referee #2:

The authors' responses clarify most points of uncertainty in the previous draft and greatly help strengthen the paper. Beyond including some of the interesting work that the authors did for the response, the one remaining point that I feel needs to be addressed is an explicit discussion of alternate hypotheses, other sources of behavioral bias, and why the authors consider the contraction hypothesis the best explanation for their results given other possibilities. These data will make a nice contribution to the literature.

1) The response to comment (1) provides an interesting analysis, and the caveat added to the interpretation is appreciated. However, it addresses a slightly different point from the original comment, which asked whether the probability of decoding each class would be affected by the previous trial in the test dataset, as you might expect if the representation of the current stimulus was shifted toward on the previous trial. The existing results are interesting, and I do not believe it is necessary to add further analyses. However, I am not quite convinced that the interpretation of neural data is in favor of a contractive effect. This is difference in opinion though, and not one that should hold up the publication of these data, so it is important to discuss the alternate explanations for neural data raised by us and reviewer 3.

Both we and reviewer 3 mentioned alternate possible hypotheses for behavioral and neural data, including anchoring, salience, signal-to-noise effects, and forgetting/confusion of past stimuli. While we understand from the responses that the authors believe a contraction bias is the most likely explanation, it is important to acknowledge and consider these alternative hypotheses in the paper and discuss why they favor the contraction hypothesis based on their current work and previous research.

2) Figures R1 and R2 provide very useful context for readers interpreting the results and relating them to other work and I believe should be included in the paper.

a) Some clarification of the R2 legend would be helpful:

- The center panel will be easier to interpret if the legend notes that the numeric values (not the color bar) show the choice accuracy for each trial type.

- In the right panel, it is difficult to parse what is being plotted. It would perhaps be easier to read if the y-axis was labeled as $P(\text{choice } S1 | \text{error})$ rather than $P(S2 > S1 | \text{error})$ and if the legend provided more detail.

b) The right panel in Fig. R2 reveals other important parts of behavior that I think should be included in the discussion along with point #1. In addition to a contractive effect, the overall error pattern in Fig. R2-right indicates that choices tend to be biased toward S1 when both stimuli are low and toward S2 when both stimuli are high value. This suggests additional sources of behavioral bias, such as lower resolution memory for S1 or decision-making that is more sensitive to the value of S2 overall. This observation is potentially relevant to other interpretations of the behavioral and neural results.

Referee #4:

Benozzo et al study history biases in ensembles of monkey PFC neurons. They find solid evidence for the attraction of S1 to S2 in the previous trial both in behavior and interestingly, early on after S1 stimulus presentation when decoding stimuli from PFC neural activity. In contrast, in later delay epochs codes do not show biases towards S2. The authors plot behavioral data and perform decoding analysis and an HMM-based analysis.

Major comments:

It is confusing that the authors say they are studying contraction bias but the behavioral figures suggest that they are indeed studying a 1-back history bias towards S2_prev. That being said, the history bias terminology is often confusing and inconsistent. While authors argue in the response to reviewers that Boboneva et al. link the two effects, this finding is based on a very specific task and circuit model with specific assumptions and by no means should be interpreted as history and contraction biases as one and the same. Therefore I agree with reviewer 2 that a logistic regression model should be included to unequivocally understand how the authors think about the origin of the effect.

How do authors interpret the fact that S1 is biased early on after stimulus presentation but then the bias gradually vanishes? Models of PPC (eg Boboneva et al) or PFC activity (eg Kilpatrick 2018 Sci Rep, Barbosa et al 2020 Nat Neurosci) usually show biases to increase over time, not to decrease. This finding should be discussed.

The motivation for the HMM analysis is unclear: Decoding could also be performed at the single-session level?

I'm afraid that the HMM doesn't add anything to the paper. It's a fancy analysis that is however not suited to reach a better understanding of the data. We learn that different neural activity profiles across ensembles characterize different periods of a trial. This is expected and doesn't seem to provide further insights, e.g. by using these decoding states for a posteriori analyses that elucidate findings beyond time-resolved decoding. Another highlighted finding of this figure is that different ensemble activity patterns are observed for red vs blue S1. This is also shown in the next figure (decoding analysis), where the finding is clearer and allows for a proper statistical assessment of the effect. I strongly suggest to completely remove the HMM analysis from the main text. The panels with "coding states" are also confusing as they imply mixed coding as a mechanism, without the finding being discussed or further elucidated.

What is a coding state? There's only a verbal description: A state that codes for a specific task variable. How was that determined? With a decoder? If the HMM ultimately is kept in the manuscript, this should be made clear

To me, decoding of prev. S2 seems like a central result and might relate to the observation of better S1 decoding in earlier figures. I.m.o., this figure should be included much earlier in the manuscript and not only after the less relevant color decoding.

I don't think that an absence of S2 decoding should be termed "activity-silent" (which would imply neurons are firing at their baseline firing rate), as the authors do in the discussion: In fact, we know from previous figures that the neurons recorded in PFC during this "activity-silent" period are in fact not silent at all, but encoding S1. So instead of an activity-silent period, it is likely we cannot decode S2 because S1 coding dominates activity.

Minor:

Figure 1: symbols are messy - maybe use dots of varying sizes with increasing sizes the further away from the center?

Methods don't describe how colors of S1 and S2 are randomized. This seems important: Are diff_color trials equally likely to same_color trials? I.e. is a change in color expected or a surprise?

Decoders trained on same color-trials seem to generalize well to trials with different colors, but not the opposite, as seen from the cross-temporal decoding plots in. There seems to be an additional signal in different-color trials that dominates activity. This could be discussed or further investigated, given that the authors focus quite a lot on the same/different distinction.

Why not use a continuous-valued regression model to decode magnitude (a continuous-valued variable) and show a quantitative bias in decoding of S1 and how it evolves over time?

END OF COMMENTS

REQUIRED ITEMS

1) - You must start the Methods section with a paragraph headed Ethical approval (https://jp.msubmit.net/cgi-bin/main.plex?form_type=display_requirements#methods).

Research must comply with The Journal's policies regarding animal experiments (<https://physoc.onlinelibrary.wiley.com/hub/animal-experiments>) and adherence to these policies must be stated in the manuscript.

Authors should confirm in their Methods section that their experiments were carried out according to the guidelines laid down by their institution's animal welfare committee, including an ethics approval reference number. The Methods section must contain a statement about access to food, water and housing, details of the anaesthetic regime: anaesthetic used, dose and route of administration, and method of killing the experimental animals.

We had reported in the previous version of the manuscript the information about the study approval in the ethical session: "by the National Institute of Mental Health Animal Care and Use Committee [LSN_03_05]".

Specifically, please include:

- Details of housing/ environment (light/dark cycle, temperature)
- The agent, route and dose of anaesthesia
- More details on termination/ euthanasia.

We wrote this sentence in the manuscript to clarify that it is an article on a previously published dataset: "*The current study is a data analysis study based on experimental recordings reported in previously published works.*" (line 135).

We do not have access to the housing/ environment (light/dark cycle, temperature) and to veterinary procedures during euthanasia and cannot be extracted by the previous articles that used the same dataset. We added this statement in the *Surgery and Histological Analysis* subsection, *Methods* section: "*The entire euthanasia procedure was conducted under extremely deep anaesthesia and under the supervision of the on-site veterinary staff*".

We have added for the euthanasia only that the perfusion was: "with 10% formol saline."

Similarly, we do not have access to information on the light/dark cycle or temperature, as these settings of the animal facility were handled by the animal care takers in compliance with the NIH regulations and not reported in the previous studies on the same dataset.

2) - Papers must comply with the Statistics Policy: https://jp.msubmit.net/cgi-bin/main.plex?form_type=display_requirements#statistics.

In summary:

- If $n \leq 30$, all data points must be plotted in the figure in a way that reveals their range and distribution. A bar graph with data points overlaid, a box and whisker plot or a violin plot (preferably with data points included) are acceptable formats.
- If $n > 30$, then the entire raw dataset must be made available either as supporting information, or hosted on a not-for-profit repository, e.g. FigShare, with access details provided in the manuscript.
- 'n' clearly defined (e.g. x cells from y slices in z animals) in the Methods. Authors should be mindful of pseudoreplication.
- All relevant 'n' values must be clearly stated in the main text, figures and tables.
- The most appropriate summary statistic (e.g. mean or median and standard deviation) must be used. Standard Error of the Mean (SEM) alone is not permitted.
- Exact p values must be stated. Authors must not use 'greater than' or 'less than'. Exact p values must be stated to three significant figures even when 'no statistical significance' is claimed.
 $p < 0.05$ is not acceptable, exact p values must be used.

We have added exact p-values as required.

EDITOR COMMENTS

Thank you for your thorough and thoughtful revision. Both reviewers agreed that the revised manuscript is much improved and that the data are of interest. Key points raised include the need to clarify the interpretation of the behavioural effect (e.g., contraction bias vs. 1-back influence), provide a more balanced discussion of alternative hypotheses, and reconsider the relevance of the HMM analysis.

We thank the reviewers for their insightful comments, which helped us clarify several aspects of the manuscript. We believe that the new logistic regression analysis better illustrates the influence of the previous S2, in addition to other factors such as the contraction bias arising from the average magnitude of the stimulus distribution. We have also expanded the discussion to address other possible sources of error, although, based on our experimental design, we do not expect additional specific biases to play a role.

We had previously explained our rationale for retaining the HMM analysis and again we emphasized here why we believe it remains relevant and does not negatively affect the interpretation of the results. However, if the reviewer remains strongly opposed to its inclusion, we are willing to remove this analysis in the next revision, because the main conclusions of the paper would remain supported without it.

Senior Editor:

Please refer to the Statistical Policy of the Journal in respect of the use of asterisks on figures to indicate "statistically significant" values. For example, in the legend to figure 4, it is written "ANOVA test with Tukey's multiple comparison test".

In this regard also, please provide the test statistic (e.g., F ratio, Mann-Whitney U) and associated degrees of freedom in all cases (i.e., not just the exact p value).

Done, removed asterisks and added test statistic values.

REFEREE COMMENTS

Referee #2:

The authors' responses clarify most points of uncertainty in the previous draft and greatly help strengthen the paper. Beyond including some of the interesting work that the authors did for the response, the one remaining point that I feel needs to be addressed is an explicit discussion of alternate hypotheses, other sources of behavioral bias, and why the authors consider the contraction hypothesis the best explanation for their results given other possibilities. These data will make a nice contribution to the literature.

1) The response to comment (1) provides an interesting analysis, and the caveat added to the interpretation is appreciated. However, it addresses a slightly different point from the original comment, which asked whether the probability of decoding each class would be affected by the previous trial in the test dataset, as you might expect if the representation of the current stimulus was shifted toward on the previous trial. The existing results are interesting, and I do not believe it is necessary to add further analyses. However, I am not quite convinced that the interpretation of neural data is in favor of a contractive effect. This is difference in opinion though, and not one that should hold up the publication of these data, so it is important to discuss the alternate explanations for neural data raised by us and reviewer 3.

Both we and reviewer 3 mentioned alternate possible hypotheses for behavioral and neural data, including anchoring, salience, signal-to-noise effects, and forgetting/confusion of past stimuli. While we understand from the responses that the authors believe a contraction bias is the most likely explanation, it is important to acknowledge and consider these alternative hypotheses in the paper and discuss why they favor the contraction hypothesis based on their current work and previous research.

We have now added a logistic regression (Fig. 2g) that considers multiple sources of bias influencing performance. However, other factors might also play a role in determining performance. One possibility is the exploration of alternative strategies, which can vary from session to session and are difficult to capture. In a previous paper (Genovesio et al., 2005), we studied strategies such as repeat–stay and change–shift, which monkeys in earlier studies adopted spontaneously to speed up the learning of stimulus–response (S–R) associations (Bussey et al., 2001). In our task, no alternative strategy could be used to

improve performance, although it is possible that the monkeys explored certain strategies in some sessions. The monkeys could not even use a lose–shift strategy to improve performance because there were no correction trials following errors. At times, monkeys may have become distracted and failed to attend to the stimulus distance; in such cases, their decisions would resemble those made in a free-choice task. In a previous study (Marcos et al., 2016), we found that a small imbalance in the activity of directional cells tuned to different target positions could bias the subsequent choice under free-choice conditions. Regarding stimulus salience, the red and blue stimuli did not differ in salience, for example, in terms of brightness level or social versus non-social relevance. The target choice was made by moving the hand from a central switch to a corresponding right or left switch, avoiding the use of a touchscreen, which in some cases might introduce biases if one movement were easier than another.

Anchoring effects, in our view, are unlikely to occur given the features of our task, considering that anchoring refers to paradigms in which an initial reference value is established and later affects judgments. Here we did not introduce such a possible reference.

We added in the discussion this part: *“Although the contraction bias effect and the effect of the last stimulus could explain a great part of the performance, other factors could have affected the performance. One possibility is the exploration of alternative strategies (Bussey et al., 2001; Genovesio et al., 2005; Fascianelli et al., 2024), but in our task, no alternative strategy could be used to improve performance, although it is possible that the monkeys explored certain strategies in some sessions. At times, monkeys may have become distracted and failed to attend to the stimulus distance; in such cases, their decisions would resemble those made in a free-choice task. In a previous study (Marcos et al., 2016), we found that a small imbalance in the activity of directional cells tuned to different target positions could bias the subsequent choice under free-choice conditions.”* (line 816)

We also added in the discussion this sentence: *“Although our analysis does not provide a mechanistic explanation, the effect of the previous S2 on the representation of S1 suggests that biases generated from the previous trial can influence the sensory representation of the current stimulus, and not only the decision process, as already shown in our previous work (Benozzo et al., 2023).”* (line 703)

2) Figures R1 and R2 provide very useful context for readers interpreting the results and relating them to other work and I believe should be included in the paper.

a) Some clarification of the R2 legend would be helpful:

- The center panel will be easier to interpret if the legend notes that the numeric values (not the color bar) show the choice accuracy for each trial type.

- In the right panel, it is difficult to parse what is being plotted. It would perhaps be easier to read if the y-axis was labeled as $P(\text{choice } S1 | \text{error})$ rather than $P(S2 > S1 | \text{error})$ and if the legend provided more detail.

These panels are part of our previous work (Figs. 2 and S1 in Benozzo et al. 2023) on how the contraction bias effect relates to both behavioral data (based only on current trial features) and neural data (during S2 presentation).

It is important to note that $P(\text{choice S1}|\text{error})$ and $P(\text{S2}>\text{S1}|\text{error})$ are not equivalent, as $\text{S2}>\text{S1}$ is independent of the animal's choice.

b) The right panel in Fig. R2 reveals other important parts of behavior that I think should be included in the discussion along with point #1. In addition to a contractive effect, the overall error pattern in Fig. R2-right indicates that choices tend to be biased toward S1 when both stimuli are low and toward S2 when both stimuli are high value. This suggests additional sources of behavioral bias, such as lower resolution memory for S1 or decision-making that is more sensitive to the value of S2 overall. This observation is potentially relevant to other interpretations of the behavioral and neural results.

We thank the reviewer for pointing out this aspect. The observation that choices are biased toward S1 for small stimuli and towards S2 for large stimuli is consistent with the contraction bias effect (see Fig.R2, left panel in the first response file). Specifically, the contraction bias favors the choice of S1 when stimuli are small, and the choice of S2 when stimuli are large.

However, we agree that other plausible explanations for this effect might exist as we discussed answering a previous question. In any case, explaining the overall behavioral performance is beyond the scope of this work. Our interest is specifically on the previous-trial effect (Fig.2a), which is computed after factoring out the average accuracy of each stimulus pair, so as to isolate only the component that depends on the previous S2 magnitude. The contraction bias is then proposed as a possible explanation for the negative slope observed in Fig.2a, as this is in line with the hypothesis that the perception of S1 is biased toward S2_pre, which in turn modulates the contraction of the current S2 when compared with S1 (i.e., the altered representation of S1). We do not claim that contraction bias is the sole source of error underlying the overall performance as we now make more clear in the revised version but we have not detected others so far that apply to other results. (added in line 816)

Referee #4:

Benozzo et al study history biases in ensembles of monkey PFC neurons. They find solid evidence for the attraction of S1 to S2 in the previous trial both in behavior and interestingly, early on after S1 stimulus presentation when decoding stimuli from PFC neural activity. In contrast, in later delay epochs codes do not show biases towards S2. The authors plot behavioral data and perform decoding analysis and an HMM-based analysis.

Major comments:

It is confusing that the authors say they are studying contraction bias but the behavioral figures suggest that they are indeed studying a 1-back history bias towards S2_prev. That being said, the history bias terminology is often confusing and inconsistent. While authors argue in the response to reviewers that Boboneva et al. link the two effects, this finding is based on a very specific task and circuit model with specific assumptions and by no means

should be interpreted as history and contraction biases as one and the same. Therefore I agree with reviewer 2 that a logistic regression model should be included to unequivocally understand how the authors think about the origin of the effect.

Thank you for pointing out this aspect, probably we put too much emphasis on the contraction bias effect, whereas our main focus is the 1-back history bias, as correctly said by the reviewer. However, the negative slope that emerged from the 1-back history bias in Fig.2a, together with our previous work on the contraction bias in the same dataset, suggests that the contraction bias is a suitable framework for interpreting our 1-back history effect.

The rationale for linking sensory history with contraction bias is as follows: previous trial history affects working memory (Visscher et al., 2009; Akrami et al., 2018; Hajonides et al., 2023), and contraction bias is a tendency for estimates in working memory to shift toward the mean of the stimulus distribution built from previously experienced stimuli. Our decision to focus on the relationship between the previous trial effect, the contraction bias, and the current trial performance, is motivated by two main factors. First, our previous work (Benozzo et al., 2023) showed that the contraction bias effect plays a predominant role, compared to other sources of bias like stimulus distance and ratio, in explaining current trial performance of this dataset. Second, this focus is further supported by the results shown in Fig.2a, where a small S2 previous tends to favour S2 selection, consistent with the hypothesis that this effect arises from a modulation of the mean stimulus distribution, as suggested by similar analyses in Akrami et al. (2018). It is important to note that this effect alters the mean performance of each stimulus pair by approximately +/- 4% (other sources of bias or error underlying the overall performance are beyond the scope of this work).

Regarding the logistic regression model, we firstly report the model used in Benozzo et al. (2023), which included only current trial regressors, i.e. the stimulus ratio and the contraction bias effect (the latter measured as the distance of S1 from the true mean stimulus distribution, i.e. 28mm). We then extended the model by adding the magnitude of S2 and S1 from the previous trial.

Fig R1. A generalised linear model (binomial distribution and logit link function) was used to predict the probability of choosing S2, across stimulus pairs with a relative distance of 8mm (10 pairs), and their related previous trial pairs (30 previous pairs each current pair). Four combinations of regressors were tested. Two models included only current trial features: one with the stimulus ratio (green), and another with the stimulus ratio plus the contraction bias effect (blue). In the other two models, we added the magnitude of S2pre (red), and additionally the magnitude of S1pre (yellow). The left panel shows the distribution of model coefficients for each regressor across models, while the right panel shows the r^2 metric for each regression model computed on the training and testing datasets. Each model was fitted on 300 randomly sampled sessions and tested on another 300 sessions. This procedure was repeated 200 times.

Models based only on current trial features (green and blue) showed a marked improvement when the contraction bias (CB) regressor was included (r^2 from ~ 0.4 to ~ 0.9). Further improvement was observed when the magnitude of the previous S2 (red) was added, allowing r^2 to exceed 0.9, whereas including the previous S1 magnitude (orange) did not produce a significant effect. It is worth noting that, although significant, the improvement provided by the previous S2 is very small, consistent with the fact that the S2-ward bias induced by the previous S2 (Fig. 2a) is on the order of only a few percentage points. Added in the manuscript, line 413, and in the Discussion line 686.

How do authors interpret the fact that S1 is biased early on after stimulus presentation but then the bias gradually vanishes? Models of PPC (eg Boboneva et al) or PFC activity (eg Kilpatrick 2018 Sci Rep, Barbosa et al 2020 Nat Neurosci) usually show biases to increase over time, not to decrease. This finding should be discussed.

As we discussed, the duration of the “silent activity” is in line at least with the results of the Ranieri et al. paper published in J. Neurosci in 2022. The reactivation of the silent trace of the previous response occurred around 700 ms after the presentation of the current visual stimulus. Differences between tasks could probably explain the differences, but it is hard to explain it based on the results of only a few studies that differ for the tasks used, and modelling studies are needed comparing different tasks.

We added in the discussion this part before our hypothesis on the timing of the reactivation in our study: “*While the timing of the reactivation of previous-trial information in our study aligns with the findings of Ranieri et al. (2022), it differs from Barbosa et al. (2020), where reactivation occurred right before stimulus presentation, together with the ramping up of neural activity. Why information is kept in an 'activity-silent' state in some cases but not in others remains unclear and might depend specifically on the task structure and task requirement*” (line 776).

The motivation for the HMM analysis is unclear: Decoding could also be performed at the single-session level? I'm afraid that the HMM doesn't add anything to the paper. It's a fancy analysis that is however not suited to reach a better understanding of the data. We learn that different neural activity profiles across ensembles characterize different periods of a trial. This is expected and doesn't seem to provide further insights, e.g. by using these decoding states for a posteriori analyses that elucidate findings beyond time-resolved decoding. Another highlighted finding of this figure is that different ensemble activity patterns are observed for red vs blue S1. This is also shown in the next figure (decoding analysis), where the finding is clearer and allows for a proper statistical assessment of the effect. I strongly suggest to completely remove the HMM analysis from the main text. The panels with "coding states" are also confusing as they imply mixed coding as a mechanism, without the finding being discussed or further elucidated.

The HMM analysis was intended to support the decoding results of the color condition (same/diffColor) by studying the presence of coding states associated with this variable. Unlike the decoding shown in Fig.6(a-c) which is a population-level analysis performed on a pseudo-dataset constructed by combining data across sessions, the HMM analysis is more localized, operating at the level of individual sessions (lines 570). This was feasible because

the color condition is a binary variable, which avoids the need to divide trials into multiple classes. As a result, it provided a good compromise between the number of available trials per session and the optimal working condition of the HMM analysis.

What is a coding state? There's only a verbal description: A state that codes for a specific task variable. How was that determined? With a decoder? If the HMM ultimately is kept in the manuscript, this should be made clear

The definition of a coding states is provided in the Methods section "*After estimating the state sequence, the presence of a coding state - defined as a state that codes for a specific task variable - was evaluated by comparing the mean state occupancy between subsets of trials grouped according to the task variable under consideration*" (line 315)

To me, decoding of prev. S2 seems like a central result and might relate to the observation of better S1 decoding in earlier figures. I.m.o., this figure should be included much earlier in the manuscript and not only after the less relevant color decoding.

That's a good point, however this result is not so robust enough to be the central focus of the present work. Indeed, we observed a loss of significance when splitting trials as reported in the previous response file (figR4) and in the main manuscript: "*However, no significant results were observed in either the match or mismatch conditions, the related curves indicated a reduction toward chance level and a shift of the accuracy peak to the early S1 presentation window, respectively.*" (line 672)

I don't think that an absence of S2 decoding should be termed "activity-silent" (which would imply neurons are firing at their baseline firing rate), as the authors do in the discussion: In fact, we know from previous figures that the neurons recorded in PFC during this "activity-silent" period are in fact not silent at all, but encoding S1. So instead of an activity-silent period, it is likely we cannot decode S2 because S1 coding dominates activity.

We thank the reviewer for raising this important point that could mislead the readers. We agree that the term "*activity-silent*" can be misleading if not explicitly defined. In our discussion, we adopted the terminology introduced by Stokes (2015, *Trends in Cognitive Sciences*), where "*activity-silent*" refers to a variable-specific latent coding state rather than to neurons firing strictly at baseline. Its use has since been adopted in several subsequent studies (e.g., Barbosa et al., 2020, and more recent work in humans).

In our data, neurons are indeed active during this period and encode S1. What we mean by "*activity-silent*", and should make explicit, is that S2 is no longer decodable from ongoing firing, i.e., its representation is silent, while S1 remains actively encoded. We will clarify this point in the revised manuscript to avoid confusion and to specify that we use "*activity-silent*" in the sense of a variable-specific loss of decodability, rather than an overall absence of neural activity.

We have added in the introduction the following sentence: *An "activity silent" state refers to the absence of an explicit coding, not to be confused with the absence of activity.*" (line 128)

Minor:

Figure 1: symbols are messy - maybe use dots of varying sizes with increasing sizes the further away from the center?

Methods don't describe how colors of S1 and S2 are randomized. This seems important: Are diff_color trials equally likely to same_color trials? I.e. is a change in color expected or a surprise?

The color of S1 and S2 were independently randomized with equal chances for the two colors for each stimulus. Consequently diff_color trials were equally likely as the same_color trials. The same randomization was performed for the right left target position of the farther stimulus.

We added in the methods: "*The presentation of the red square and blue circle of S1 and S2 were independently randomized*" (line 166).

Decoders trained on same color-trials seem to generalize well to trials with different colors, but not the opposite, as seen from the cross-temporal decoding plots in. There seems to be an additional signal in different-color trials that dominates activity. This could be discussed or further investigated, given that the authors focus quite a lot on the same/different distinction.

Yes, we agree that in Fig.3d the cross-temporal decoding shows more significant clusters than in Fig.3c. This could indeed suggest that during the second half of the S1 presentation, some activity specific to the different-color condition emerges. However, given their temporal discontinuity and the fact that this is not confirmed in the within-condition in panel c, it is difficult to speculate on their potential meaning.

Why not use a continuous-valued regression model to decode magnitude (a continuous-valued variable) and show a quantitative bias in decoding of S1 and how it evolves over time?

The small number of trials for each specific combination of variables precluded this analysis, for this reason we chose an across session decoding as the main analysis, or an HMM analysis within sessions but leveraging on the information of the neurons recorded simultaneously.

Dear Dr Genovesio,

Re: JP-RP-2025-288070R2 "History bias and its perturbation of the stimulus representation in the macaque prefrontal cortex." by Danilo Benozzo, Lorenzo Ferrucci, Francesco Ceccarelli, and Aldo Genovesio

Thank you for submitting your manuscript to The Journal of Physiology. It has been assessed by a Reviewing Editor and by 2 expert referees and we are pleased to tell you that it is acceptable for publication following satisfactory revision.

REVISION CHECKLIST:

We look forward to receiving your revised submission.

Yours sincerely,

Richard Carson
Senior Editor
The Journal of Physiology

EDITOR COMMENTS

Reviewing Editor:

The manuscript has been evaluated positively overall and is considered acceptable pending final revisions. At this stage, please focus on clarifying interpretation, improving methodological exposition, and streamlining presentation, particularly with respect to the behavioral regression model, the interpretation of decoding effects, and the description of the HMM analysis. Substantial new analyses are not expected; please ensure that claims are appropriately calibrated to the analyses presented.

Senior Editor:

Please note that there remain instances in which the test statistic (e.g., Mann Whitney U) and the associated degrees of freedom are not reported. In addition, the degrees of freedom for the outcome of an ANOVA are given as a single value. Please ensure that both the numerator and denominator degrees of freedom are given.

REFEREE COMMENTS

Referee #2:

The authors have addressed my remaining concerns.

Referee #4:

The study provides interesting analyses of an existing dataset w.r.t. history biases and their neural encoding in monkeys in a delayed comparison task. this study bridges to history biases in similar tasks in rodent literature and to primate literature in different but related tasks, guaranteeing its impact. this comparability with existing literature is seen as a positive characteristic of this study. interesting new analyses on color match/mismatch across trial are original.

Insight into physiological mechanisms are given but somewhat limited by the low number of neurons and the complex experimental design. however, given that this is primate research, we nonetheless gain valuable insights about the encoding of stimulus history in monkey prefrontal cortex. Results are shown in multiple complementary ways. For final conclusions from the study, please find detailed comments in the attachment.

END OF COMMENTS

Regression model

- 1) Benozzo et al have now added a regression models and model comparisons with reduced versions of the full model:

$$p(\text{left}) = \beta_0 + \beta_1 \cdot \text{stim ratio} + \beta_2 \cdot (S1_{\text{current}} - \langle S \rangle) + \beta_3 \cdot S2_{\text{pre}} + \beta_4 \cdot S1_{\text{pre}}$$

They compare cross-validated R2 across models. It is unclear why they mix regressor formats:

- The stim ratio is a fraction, assuming a non-linear comparison of S1 and S2,
- CB is a relative distance measure (a deviation of the current stimulus from the mean), assuming a linear effect
- S1pre and S2pre are absolute values (not relative to S1_current)

Combining ratio, relative and absolute representations makes the comparison of weights non-straightforward.

For comparison, Akrami et al. use a consistent regressor format, which allows the corresponding coefficients to be compared directly.

The authors should present a regression model with consistent coding of variables.

- 2) Relatedly, the current comparison of weights suggests that the decision to choose left is in fact more strongly driven by the CB on S1 than by the actual stimulus ratio (as also stated in Benozzo 2023). I wonder whether this result remains valid when a consistent regressor format (e.g. Akrami et al) is chosen.
- 3) Similarly, the results suggest that the CB explains more variance than bias towards S2, but a consistent choice of coding should be presented to support this claim.
- 4) Taken together, the authors should state more clearly in the results section, just before going to the neural results section, that the main effect in their behavioral results stems from CB, but that they nonetheless focus on the neural correlates of the weaker S2_prev-effect for practical reasons and reasoning in the previous literature.

Interpretation of reactivation of “activity-silent” codes

- 1) I hadn't understood that the authors interpreted the interaction effect of S2_prev x S1_current on S1 decoding as “reactivation from a silent trace”. Instead, I had read Fig 4a,c as S1 encoding being biased towards S2_prev. It is not clear from the analysis whether the decoding is the result of one versus the other.

A more straightforward analysis is the decoding of S2_prev throughout the current trial, as in Barbosa et al., and as reported in Fig 6d. I suggest moving 6d alongside Fig 4a,c so that the interpretation in terms of S2 reactivation is more clearly interpreted. Please at minimum state clearly in the results that you interpret the S1 decoding bias as a reactivation, so that the discussion is easier to follow.

Decoding states/HMM

- 2) “After estimating the state sequence, the presence of a coding state - defined as a state that codes for a specific task variable - was evaluated by comparing the mean state occupancy between subsets of trials grouped according to the task variable under consideration”

This is not a precise definition. I understand that this analysis has been used before, but

methods for the current paper should still be sufficiently tractable without having to search through the authors' publication history.

- 3) I still think the HMM analysis is overly complex for what the authors want to show: that there are neurons whose firing patterns during S1 presentation is described by firing rate = condition x color , and that these neurons don't show the same interaction effect during pre-stimulus. If the authors want to keep the HMM analysis, I suggest they streamline the HMM results description, get rid of overly complicated introductions to the technique/previous results etc. and highlight what we learn.

Taken together, I think the manuscript can be published if the above comments are addressed for interpretability and readability.

EDITOR COMMENTS

Reviewing Editor:

The manuscript has been evaluated positively overall and is considered acceptable pending final revisions. At this stage, please focus on clarifying interpretation, improving methodological exposition, and streamlining presentation, particularly with respect to the behavioral regression model, the interpretation of decoding effects, and the description of the HMM analysis. Substantial new analyses are not expected; please ensure that claims are appropriately calibrated to the analyses presented.

We have addressed the remaining reviewer's concerns. Several comments were helpful in clarifying ambiguities in our description, particularly regarding the activity-silent period, which we have revised accordingly. We have also reorganized in part the results on the HMM analysis.

In addition, we have used new regression models to address the remaining concerns and to better parallel previous work by Akrami et al. However, to avoid expanding the manuscript with additional results on this aspect, we chose to include these new analyses in the response to the reviewers rather than in the revised manuscript.

Senior Editor:

Please note that there remain instances in which the test statistic (e.g., Mann Whitney U) and the associated degrees of freedom are not reported. In addition, the degrees of freedom for the outcome of an ANOVA are given as a single value. Please ensure that both the numerator and denominator degrees of freedom are given.

Thank you; the missing test statistics and degrees of freedom have now been added.

REFEREE COMMENTS

Referee #2:

The authors have addressed my remaining concerns.

Referee #4:

The study provides interesting analyses of an existing dataset w.r.t. history biases and their neural encoding in monkeys in a delayed comparison task. This study bridges to history biases in similar tasks in rodent literature and to primate literature in different but related tasks, guaranteeing its impact. This comparability with existing literature is seen as a positive characteristic of this study. interesting new analyses on color match/mismatch across trial are original.

Insight into physiological mechanisms are given but somewhat limited by the low number of neurons and the complex experimental design. However, given that this is primate research, we nonetheless gain valuable insights about the encoding of stimulus history in monkey prefrontal cortex. Results are shown in multiple complementary ways. For final conclusions from the study, please find detailed comments in the attachment.

Regression model

1) Benozzo et al have now added a regression models and model comparisons with reduced versions of the full model: $p(\text{left}) = \beta_0 + \beta_1 \cdot \text{stim ratio} + \beta_2 \cdot (S1_{\text{current}} - \langle S \rangle) + \beta_3 \cdot S2_{\text{pre}} + \beta_4 \cdot S1_{\text{pre}}$

They compare cross-validated R2 across models. It is unclear why they mix regressor formats:

- The stim ratio is a fraction, assuming a non-linear comparison of S1 and S2,
- CB is a relative distance measure (a deviation of the current stimulus from the mean), assuming a linear effect
- S1pre and S2pre are absolute values (not relative to S1_current)

Combining ratio, relative and absolute representations makes the comparison of weights non-straightforward. For comparison, Akrami et al. use a consistent regressor format, which allows the corresponding coefficients to be compared directly. The authors should present a regression model with consistent coding of variables.

We agree with the reviewer that caution is required when interpreting regression weights. Although all regression coefficients were normalized to facilitate interpretation, we refrain from directly comparing them across regressors with different functional forms. Instead, our conclusions are based on changes in model prediction performance (lines 418-424) and, where relevant, on the robustness of coefficient signs across models (i.e. same regressor across models, in line with how weights were grouped in Fig.2g left panel).

Following the reviewer's suggestion, we fitted a regression model based exclusively on stimulus magnitudes: $P(\text{select } S2) = \beta_0 + \beta_1 \cdot S2_{\text{current}} + \beta_2 \cdot S1_{\text{current}} + \beta_3 \cdot S2_{\text{pre}} + \beta_4 \cdot S1_{\text{pre}}$ (Fig.R1). Overall, the results are consistent with those obtained using difficulty-based regressors (Fig.2g). The predictive performance is slightly reduced, due to the absence of an explicit interaction term. This is illustrated in Fig.R2, where $P(\text{select } S2) = \beta_0 + \beta_1 \cdot S2_{\text{current}} + \beta_2 \cdot S1_{\text{current}} + \beta_3 \cdot \text{ratio}$.

This model resembles the one including only the stimulus ratio and CB, since CB can be expressed as a shifted version of S1 relative to $\langle S \rangle$. Notably, the weight associated with S2current is markedly reduced, while the signs of the coefficients for S1current and the ratio are consistent with those of the CB and ratio in the blue model shown in Fig.2g. Adding S2pre improved predictions (not dramatically, but in line with the difficulty-based models), whereas adding S1pre did not further improve predictions, as its coefficient remains close to zero.

Fig R1. Generalised linear models (binomial distribution and logit link function) based on stimulus magnitudes were used to predict the probability of choosing S2, across stimulus pairs with a relative distance of 8mm (10 pairs), and their related previous trial pairs (30 previous pairs each current pair). Three regressor configurations were tested: a model including only current-trial stimuli (purple), a model additionally including the magnitude of S2pre (blue), and a model further including the magnitude of S1pre (green). The left panel shows the distribution of regression coefficients across models, while the right panel shows the r² metric for each regression model computed on the training and testing datasets. Each model was fitted on 300 randomly sampled sessions and tested on another 300 sessions. This procedure was repeated 200 times.

Figure 2R. Regression model with only current trial stimulus magnitude and an interaction term (stimulus ratio: $\text{sign}(S2-S1)\max(S1, S2)/\min(S1, S2)$). Same fitting procedure done in Fig.R1. This resembles the blue model shown in Fig.2g where as regressors we used $CB = S1 - \langle S \rangle$ and stimulus ratio.

Revised line 230 to better clarify the meaning of CB and ratio regressors.

2) Relatedly, the current comparison of weights suggests that the decision to choose left is in fact more strongly driven by the CB on S1 than by the actual stimulus ratio (as also stated in Benozzo 2023). I wonder whether this result remains valid when a consistent regressor format (e.g. Akrami et al) is chosen.

The regression models fitted in Akrami et al. were designed to account for the history of rewards, choices and stimuli, but did not include regressors explicitly encoding trial difficulty. As a consequence, we do not see a straightforward way to infer the role of the stimulus ratio solely from weights associated with individual stimulus magnitudes. Perhaps Fig.2R partially addresses this question on the role of CB and ratio vs. S1 and S2. We also highlight that several of the regressors we considered are strongly interdependent. For example: $CB = S1 - \langle S \rangle$ (when using S1 instead of CB, the contribution of $\langle S \rangle$ is absorbed into the intercept β_0), the stimulus ratio increases as stimulus magnitude decreases (because only stimulus pairs with a fixed relative distance of 8 mm were used), similarly S1 and S2: for a given S1, S2 can only take values of $S1+8$ or $S1-8$.

3) Similarly, the results suggest that the CB explains more variance than bias towards S2, but a consistent choice of coding should be presented to support this claim.

4) Taken together, the authors should state more clearly in the results section, just before going to the neural results section, that the main effect in their behavioral results stems from CB, but that they nonetheless focus on the neural correlates of the weaker S2_prev-effect for practical reasons and reasoning in the previous literature.

We agree. As shown in the right panel of Fig.2g, CB accounts for a larger fraction of explained variance than S2pre. From Results section, line 424: "It is worth noting that, although significant, the improvement provided by the previous S2 is relatively small, consistent with the fact that the S2-ward bias induced by the previous S2 is on the order of only a few percentage points (see Fig. 2a)."

We also added in the final part of the discussion "to a lesser degree" in the following sentence: "Although the contraction bias effect and to a lesser degree the effect of the last stimulus could explain a great part of the performance, other factors could have affected the performance."

Interpretation of reactivation of "activity-silent" codes

1) I hadn't understood that the authors interpreted the interaction effect of S2_prev x S1_current on S1 decoding as "reactivation from a silent trace". Instead, I had read Fig 4a,c as S1 encoding being biased towards S2_prev. It is not clear from the analysis whether the decoding is the result of one versus the other.

A more straightforward analysis is the decoding of S2_prev throughout the current trial, as in Barbosa et al., and as reported in Fig 6d. I suggest moving 6d alongside Fig 4a,c so that the interpretation in terms of S2 reactivation is more clearly interpreted. Please at minimum state clearly in the results that you interpret the S1 decoding bias as a reactivation, so that the discussion is easier to follow.

Thank you for highlighting this point, as it gave us the opportunity to clarify and rephrase statements that may have been misleading. Our interpretation of Fig 4a,c is that S1 encoding is biased towards S2pre. We do not interpret the interaction effect (S2pre x S1current) on S1 decoding as arising from a reactivation of an activity-silent trace, due to a mismatch in timing. "the reactivation of the previous S2 representation occurred primarily during the late S1 period, whereas the initial phase of S1 presentation was most influenced by the previous S2 value" (line 776).

To avoid confusion we revised line 32: "Interestingly, this effect coincided with an "activity-silent" period, followed by the reactivation of the decoding of the previous stimulus magnitude", and line 58: "The effect of the previous stimulus magnitude on the representation of the first current stimulus was stronger during the period in which the past stimulus was not explicitly decoded (the activity-silent phase), and preceded its reactivation."

Decoding states/HMM

2) "After estimating the state sequence, the presence of a coding state - defined as a state that codes for a specific task variable - was evaluated by comparing the mean state occupancy between subsets of trials grouped according to the task variable under consideration"

This is not a precise definition. I understand that this analysis has been used before, but methods for the current paper should still be sufficiently tractable without having to search through the authors' publication history.

The definition of coding state has been revised, line 321.

3) I still think the HMM analysis is overly complex for what the authors want to show: that there are neurons whose firing patterns during S1 presentation is described by firing rate = condition x color, and that these neurons don't show the same interaction effect during pre-stimulus. If the authors want to keep the HMM analysis, I suggest they streamline the HMM results description, get rid of overly complicated introductions to the technique/previous results etc. and highlight what we learn.

We have worked on the results on the HMM model cutting unnecessary parts. We understand the reviewer's concern; we also think that the HMM analysis is not essential, but at the same time we believe that it can provide further support for the results and may be interesting to some readers. Placing it at the end of the Results section is therefore in line with its more limited importance compared with the previous results.

Taken together, I think the manuscript can be published if the above comments are addressed for interpretability and readability.

Dear Professor Genovesio,

Re: JP-RP-2026-288070R3 "History bias and its perturbation of the stimulus representation in the macaque prefrontal cortex." by Danilo Benozzo, Lorenzo Ferrucci, Francesco Ceccarelli, and Aldo Genovesio

We are pleased to tell you that your paper has been accepted for publication in The Journal of Physiology.

Yours sincerely,

Richard Carson
Senior Editor
The Journal of Physiology

IMPORTANT POINTS TO NOTE FOLLOWING ACCEPTANCE OF YOUR PAPER:

- **IMPORTANT NOTICE ABOUT OPEN ACCESS:** To assist authors whose funding agencies mandate immediate public access to published research findings, The Journal of Physiology allows authors to pay an Open Access (OA) fee to have their papers made freely available immediately on publication.

- You can help your research get the attention it deserves! Check out Wiley's free Promotion Guide for best-practice recommendations for promoting your work at: www.wileyauthors.com/eoo/guide. You can learn more about Wiley Editing Services which offers professional video, design, and writing services to create shareable video abstracts, infographics, conference posters, lay summaries, and research news stories for your research at: www.wileyauthors.com/eoo/promotion.

- If you would like to receive our 'Research Roundup', a monthly newsletter highlighting the cutting-edge research published in The Physiological Society's family of journals (The Journal of Physiology, Experimental Physiology, Physiological Reports, The Journal of Nutritional Physiology and The Journal of Precision Medicine: Health and Disease), please click this link, fill in your name and email address and select 'Research Roundup': <https://www.physoc.org/journals-and-media/membernews>

EDITOR COMMENTS

Reviewing Editor:

Thank you for your careful and thorough responses to the reviewer's comments. The revisions have clarified the interpretation of the behavioural and neural analyses, improved the presentation of the regression and HMM results, and appropriately calibrated the manuscript's claims.